# AORCHESTRA: Automating Sub-Agent Creation for Agentic Orchestration

Jianhao Ruan [* 1 2]   Zhihao Xu [* 3]   Yiran Peng [1]   Fashen Ren [2]   Zhaoyang Yu [1]   Xinbing Liang [4]   Jinyu Xiang [2]   Yongru Chen [2]   Bang Liu [5]   Chenglin Wu [1]   Yuyu Luo [2]   Jiayi Zhang [1 2]

## Abstract

Language agents have shown strong promise for task automation. Realizing this promise for increasingly complex, long-horizon tasks has driven the rise of a subagent-as-tools paradigm for multi-turn task solving. However, existing designs still lack a *dynamic abstraction* view of sub-agents, thereby hurting adaptability: sub-agents are either context-isolated threads that lack specialization, or static roles that require human-engineering. We address this challenge with a unified, framework-agnostic agent abstraction that models any agent as a tuple $\langle Instruction, Context, Tools, Model \rangle$. This tuple acts as a compositional recipe for capabilities, enabling the system to spawn specialized executors for each task on demand. Building on this abstraction, we introduce an agentic system AORCHESTRA, where the central orchestrator concretizes the tuple at each step: it curates task-relevant context, selects tools and models, and delegates execution via on-the-fly automatic agent creation. Such designs enable reducing human engineering efforts, and remain framework-agnostic with plug-and-play support for diverse agents as task executors. It also enables a controllable performance–cost trade-off, allowing the system to approach Pareto-efficient. Across three challenging benchmarks (GAIA, SWE-Bench, Terminal-Bench), AORCHESTRA achieves 16.28% relative improvement against the strongest baseline when paired with Gemini-3-Flash. Our code is publicly available at https://github.com/FoundationAgents/AOrchestra.

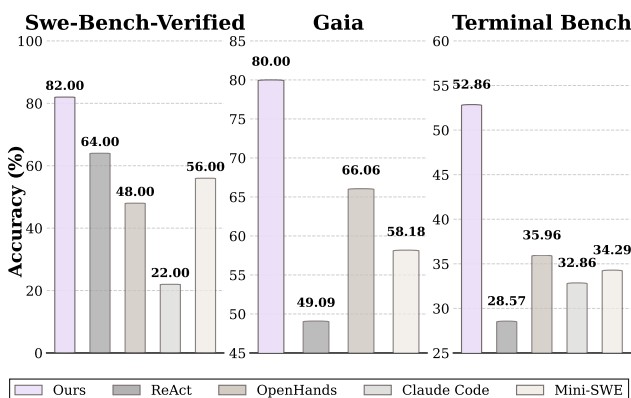

*Figure 1.* Overall performance on three challenging agentic benchmarks (GAIA, Terminal-Bench-2, SWE-Bench-Verified) paired with Gemini-3-Flash when comparing AORCHESTRA against other popular agentic frameworks.

## 1. Introduction

Humans handle complex, long-horizon work via collective intelligence and the ability to coordinate (Gao et al., 2025a; Zhu et al., 2025b). As today's agents are pushed toward similarly complex and multi-turn tasks, a well-designed agentic system becomes a vital way to scale performance beyond a single model (Liu et al., 2025b).

To cope with increasingly complex scenarios, early attempts rely on fixed coordination workflows (Zhang et al., 2024) or multi-agent systems (Hong et al., 2023; Hu et al., 2025; Li et al., 2025a). While multi-agent collaboration can improve task decomposition, in open-ended environments it often incurs substantial coordination overhead and provides limited control over context routing, leading to either noisy over-sharing or harmful omission of critical information, which makes robust long-horizon execution difficult (Gao et al., 2025b; Wu et al., 2026).

More recent approaches therefore move toward a more practical sub-agent-as-tools paradigm, where a main agent (orchestrator) delegates a task to a sub-agent via an explicit tool call. Yet existing designs still lack flexibility in practice and often degenerate into two limited patterns: (1) *Sub-agents as context-isolation threads.* Systems such as Schroeder et al. (2025); Sun et al. (2025) primarily treat sub-agents as

---
[*]Equal contribution [1]DeepWisdom [2]The Hong Kong University of Science and Technology (Guangzhou) [3]Renmin University of China [4]East China Normal University [5]Université de Montréal & Mila. Correspondence to: Yuyu Luo <yuyuluo@hkust-gz.edu.cn>, Jiayi Zhang <jzhang361@connect.hkust-gz.edu.cn>.

*Proceedings of the 43rd International Conference on Machine Learning*, Seoul, South Korea. PMLR 306, 2026. Copyright 2026 by the author(s).

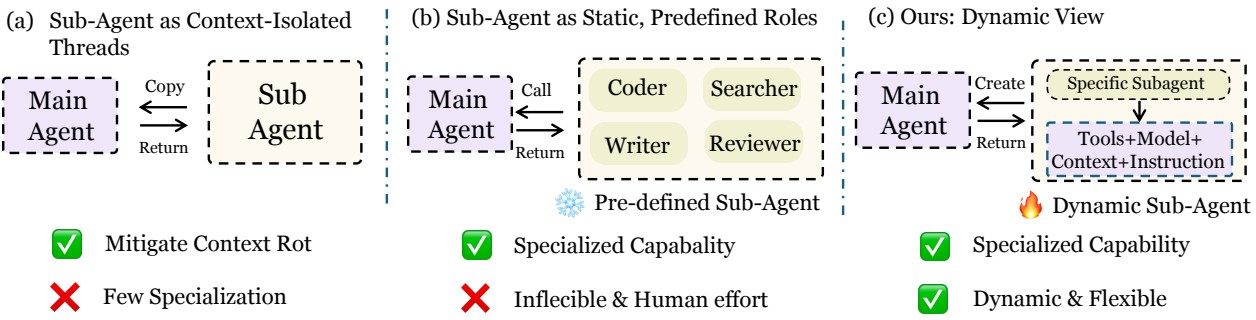

*Figure 2.* Comparison of subagent-as-tools approaches. (a) Sub-agents as context-isolated threads mitigate context rot but lack on-demand specialization. (b) Sub-agents as static roles provide specialized capabilities but are inflexible, leave coverage gaps, and require heavy human engineering. (c) Our dynamic agent orchestration concretizes a unified 4-tuple recipe (Instruction, Context, Tools, Model) to instantiate task-tailored executors on the fly, enabling on-demand specialization and task-sufficient context.

isolated context threads, aiming to prevent context rot (Hong et al., 2025). However, in real-world tasks, subtasks often require specialized capabilities. Therefore, such systems fail to fully realize the potential of specialized sub-agents. (2) *Sub-agents as static roles.* Systems such as Anthropic (2025); Li et al. (2025c) treat each sub-agent as a static role, and their capabilities or their coordination patterns are typically hard-wired. A pre-defined set of sub-agents cannot cover the dynamically emerging variety of subtasks in open environments (Zhang et al., 2026; 2025a). Besides, it relies on heavy human engineering, making the system difficult to adapt to various environments.

In this paper, we introduce AORCHESTRA, an agentic framework designed to solve long-horizon, complex tasks, to address the above challenges. Our first key idea is to treat each subagent from a dynamic view, as it is often impractical to pre-define a fixed set of sub-agents that can cover all emerging needs. We posit that a sub-agent should be treated as a *dynamically creatable* abstraction unit, allowing the system to create and instantiate the tailored sub-agent at run-time by composing its capabilities to match the current task. Concretely, *any* agent can be described as an instantiable unit via a unified four-tuple: (INSTRUCTION, CONTEXT, TOOLS, MODEL). This abstraction is organized around two complementary axes essential for an agent's task solving: (1) *Working memory (instruction, context):* what the agent must achieve and what evidence it should condition on. (2) *Capabilities (tools, model):* what the agent is empowered to do to accomplish that objective. This design and dynamic view make sub-agents freely creatable on demand.

Building on this dynamic view, we further introduce a dedicated orchestrator that operates directly over the four-tuple interface to automatically create tailored sub-agents on the fly. It does not execute any tasks and focuses exclusively on orchestration, where we define it as dynamically decomposing the overall objective into the next subtask, creating and

delegating a specialized tailed sub-agent for task execution via explicit tool calls. This decoupling design offers several key advantages. First, this dynamic creation allows each subagent to be customized with unique capabilities and a clean working context, significantly improving task execution accuracy. Second, the orchestrator remains agnostic to the internal implementation of sub-agents, making them fully pluggable. Third, the orchestrator can be trained or learned from interactive experience. This ranges from basic skills for agent creation to advanced features like adaptive model selection, achieving an optimal balance between cost and performance.

Through extensive experiments, we demonstrate AORCHESTRA achieves stronger performance and broader generalization in open-world settings. We first evaluate our framework in a training-free setting on three challenging agentic benchmarks: Terminal-Bench 2.0 (Team, 2025) (bash environment), SWE-Bench (Jimenez et al., 2023) (coding environment), and GAIA (Mialon et al., 2023) (digital world environment). Across all benchmarks, our method consistently outperforms both representative sub-agent orchestration approaches (Anthropic, 2025) and widely used agent frameworks (Wang et al., 2024; Yang et al., 2024). In particular, our framework achieves a 16.28% improvement when paired with `Gemini3-Flash`, validating the superiority of our orchestration model in complex, long-horizon tasks. Importantly, AORCHESTRA naturally supports learning the orchestration policy from experience. We instantiate this in two ways: (1) we apply supervised fine-tuning to improve the Orchestrator's subtask decomposition and 4-tuple synthesis, leading to better orchestration quality by +11.51% pass@1 on GAIA and (2) we leverage in-context learning to optimize cost-aware routing, which improves GAIA pass@1 by +3.03% while reducing average cost by 18.5%, resulting in a more favorable cost–performance Pareto frontier.

Overall, our contributions are:

- We propose AORCHESTRA, an orchestrator-centric agentic system that treats sub-agents as *dynamically creatable* executors via a unified 4-tuple interface (INSTRUCTION, CONTEXT, TOOLS, MODEL), enabling on-demand specialization with task-sufficient context and explicit capability control.

- AORCHESTRA achieves strong training-free performance on Terminal-Bench 2.0, SWE-Bench-Verified, and GAIA, consistently outperforming popular agentic systems. We achieve 16.28% relative improvement against the strongest baseline when paired with Gemini-3-Flash.

- We show the orchestration policy is learnable under this design from two complementary angles: supervised fine-tuning improves subtask decomposition and 4-tuple synthesis, and cost-aware routing via in-context learning yields favorable cost–performance Pareto trade-offs.

## 2. Related Work

**Multi-Agent Systems**  Inspired by collaborative problem solving, early efforts propose multi-agent systems (MAS) to enhance the task-solving capability of language models (Zhang et al., 2025b; Wu et al., 2024; Shi et al., 2025b; Gao et al., 2025b; Zhu et al., 2025a; Fang et al., 2025; Chen et al., 2023; Li et al., 2025b). For example, MetaGPT (Hong et al., 2023) organizes agents into a structured software-development workflow, where specialized roles (e.g., product manager, architect) collaborate via predefined communication protocols. OWL (Hu et al., 2025) adopts a planner-worker workflow to improve transfer and generalization by modularizing domain-agnostic planning and domain-specific execution. Despite their effectiveness, most MAS typically rely on a fixed workflow, leading to rigidity. Although AutoAgents (Chen et al., 2023) proposes building different multi-agent systems for each task, they still rely on a fixed workflow to accomplish this. This motivates a growing shift toward the *sub-agents-as-tools* paradigm, and we will list related works in the next part (Gao et al., 2025a;b). AORCHESTRA follows the latter and further emphasizes orchestration-centric, dynamic sub-agent creation without relying on a specific human-designed workflow.

**Sub-Agent as Tools**  This approach involves a primary agentic model invoking a sub-agent in a tool-like manner to solve problems (Li et al., 2025c; Su et al., 2025; Grand et al., 2025). For example, THREAD (Schroeder et al., 2025) enables the recursive spawning of sub-agents to address decomposed subproblems. Similarly, Context-Folding (Sun et al., 2025) proposes branching for a subtask and then folding it back by compressing intermediate steps into a concise summary, thereby managing context. However, these methods do not treat sub-agents as fully specialized agents, leading to their insufficient utilization. Other practical systems, such as Claude Code (Anthropic, 2025), support sub-agents that operate within isolated context windows with custom system prompts and tool permissions. Yet, these sub-agents are typically configured as fixed specialists and still require manual design. AORCHESTRA addresses these limitations by treating each sub-agent as a dynamic unit and proposes an orchestration-centric agentic system that proactively and dynamically creates such sub-agents on demand.

## 3. Methodology

### 3.1. Problem Formulation

In this paper, we mainly focus on solving complex agentic tasks. The agentic system solve a user goal $G$ through multi-step interaction with an environment. The environment exposes an *environment-level* action space $\mathcal{A}_{\text{env}}$ (e.g., shell commands, web operations, code edits) and returns feedback such as observations, tool outputs, and error messages. An interaction trajectory for a task can be therefore defined as:

$$\tau = (s_0, a_0, o_0, s_1, a_1, o_1, \ldots, s_T),$$

where $s_t \in \mathcal{S}$ denotes the system state at step $t$ (including accumulated history, intermediate results, and environment feedback), $a_t$ is the action taken at step $t$, and $o_t \in \mathcal{O}$ is the returned observation. The system evolves according to a state-transition function

$$s_{t+1} = \delta(s_t, a_t, o_t),$$

where $\delta : \mathcal{S} \times \mathcal{A}_{\text{env}} \times \mathcal{O} \to \mathcal{S}$ maps the current state, action, and observation to the next state by incorporating newly returned information into the system's internal state.

**Sub-agent-as-tools view.**  We focus on the sub-agent-as-tools paradigm, where a *main agent* (orchestrator) can either act in the environment directly or delegate a subtask to a sub-agent as a tool call. Accordingly, the orchestrator operates over a *system-level* action space that typically includes three types of actions: (1) environment actions $u \in \mathcal{A}_{\text{env}}$, (2) delegation actions ($\texttt{Delegate}(\cdot)$) that invoke a sub-agent to execute, and (iii) termination ($\texttt{Finish}$). We denote this generic orchestration action space as

$$\mathcal{A}_{\text{orch}} \supseteq \mathcal{A}_{\text{env}} \cup \{\texttt{Delegate}(\cdot), \texttt{Finish}(y)\}.$$

Different systems mainly differ in how $\texttt{Delegate}$ is parameterized (e.g., delegating with only context vs. delegating to a fixed set of roles), and in whether the orchestrator itself also performs environment actions.

Our objective is to maximize task success, optionally trading off execution cost:

$$\max_{\pi} \mathbb{E}\Big[\mathbf{1}\{\text{Success}(G)\} - \lambda \cdot \text{Cost}(\tau)\Big],$$

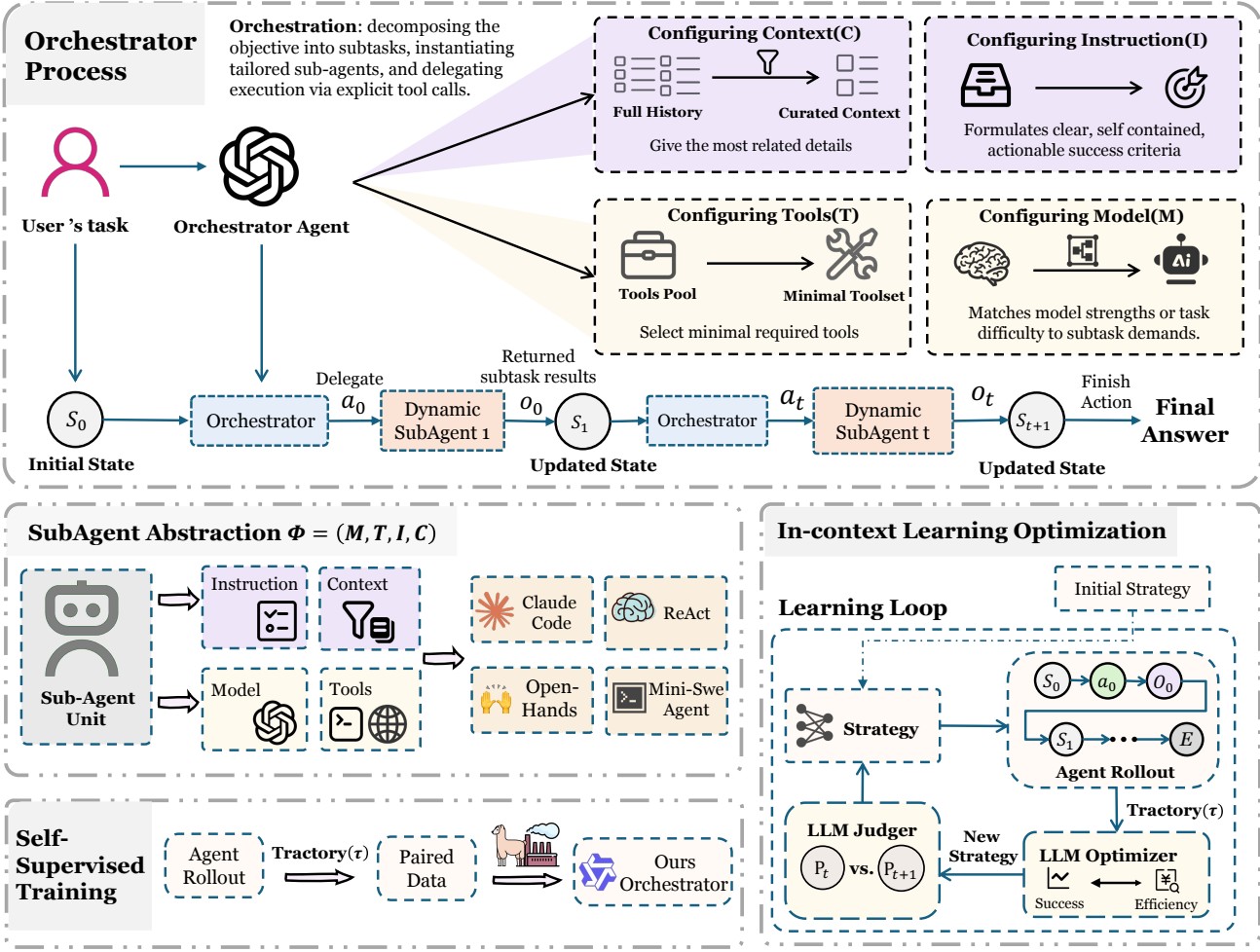

*Figure 3.* Overall design of our proposed agentic framework, AORCHESTRA, for complex, long-horizon tasks. The orchestrator solves a user task by repeatedly delegating subtasks to on-the-fly instantiated sub-agents, each defined by a unified four-tuple $(I, C, T, M)$. The orchestrator is learnable and can improve its decomposition, context routing, and capability allocation from past experience.

where $\pi$ is the orchestrator policy, $\text{Cost}(\tau)$ may include token usage, tool calls, latency, or monetary cost, and $\lambda$ controls the cost–performance trade-off.

### 3.2. AORCHESTRA

**A unified four-tuple agent abstraction.** AORCHESTRA models *both* the main agent and sub-agents under a unified framework-agnostic abstraction. We define an agent instance as an instantiable four-tuple

$$\Phi = (I, C, T, M),$$

where $I$ is the task instruction specifying the current objective and success criteria, $C$ is the curated working context the agent conditions on, $T$ is the tool set defining the agent's action space, and $M$ is the underlying model that interacts with the environment. This abstraction explicitly separates two complementary axes that require *specialization*: *working memory* $(I, C)$ and *capabilities* $(T, M)$. Notably, we

do not view a sub-agent as a static entity, but rather as a *dynamic* unit that can be parametrized and created at runtime.

The main agent (orchestrator) can also be represented by a tuple $\Phi^{\text{main}} = (I^{\text{main}}, C^{\text{main}}, T^{\text{main}}, M^{\text{main}})$. The difference is that $T^{\text{main}}$ exposes *system tools* for orchestration (e.g., `Delegate`, `Finish`) rather than environment tools in $\mathcal{A}_{\text{env}}$.

**Action Space of Orchestrator.** Building on this abstraction, AORCHESTRA decouples orchestration from execution. The orchestrator in AORCHESTRA never directly takes environment actions in $\mathcal{A}_{\text{env}}$. Instead, it operates only the two following actions:

$$\mathcal{A}_{\text{AORCHESTRA}} = \{\texttt{Delegate}(\Phi), \texttt{Finish}(y)\}.$$

At step $t$, the orchestrator samples an action $a_t \in \mathcal{A}_{\text{AORCHESTRA}}$. If $a_t = \texttt{Delegate}(\Phi_t)$, it spawns an executor $A(\Phi_t)$ to execute the subtask and returns an observation

$o_t$. If $a_t = \text{Finish}(y)$, the interaction terminates with the final answer $y$. Returned observations are integrated into the next state via $s_{t+1} = \delta(s_t, a_t, o_t)$.

**Implementation of `Delegate` and `Finish`.** We implement AORCHESTRA with two system tools available to the Orchestrator: `Delegate` and `Finish`. `Delegate` takes $\Phi_t = (I_t, C_t, T_t, M_t)$ as arguments and instantiates an executor accordingly. The executor runs with model $M_t$, is restricted to the tool set $T_t$, and conditions only on $(I_t, C_t)$. It returns a structured observation $o_t$ to the Orchestrator, typically including (i) a concise result summary, (ii) relevant artifacts (e.g., files, references), and (iii) error messages or logs if execution fails. `Finish` terminates the interaction and outputs the final response $y$.

**Advantages of AORCHESTRA** Our proposed AORCHESTRA offers several key advantages. First, it dynamically equips each sub-agent with tailored capabilities on demand, which substantially improves the accuracy of task execution. Unlike prior works (Sun et al., 2025; Anthropic, 2025), the orchestrator deliberately provides well-structured context for the sub-agent to use. As shown later in Section 4.3.1, this careful context management enhances the model's ability to solve tasks. Second, the orchestrator operates solely on a four-tuple abstraction and remains independent of the internal implementation of sub-agents. This flexibility allows us to employ various designs for sub-agents, such as a simple React approach (Yao et al., 2022) or a mini-SWE agent. Third, the orchestrator can learn from extensive experience. We will then detail this in Section 3.3. These learnable aspects include basic task orchestration skills (i.e., what to do, what to condition on, and which tool to use) as well as advanced features (e.g., adaptive model routing, where the goal might be to balance performance and cost by selecting the most suitable model).

### 3.3. Learnable Orchestrator

With $\mathcal{A}_{\text{AORCHESTRA}} = \text{Delegate}(\Phi), \text{Finish}(y)$, the orchestration task can be expressed as learning a policy over structured actions:

$$\pi_\theta(a_t \mid s_t), \quad a_t \in \mathcal{A}_{\text{AORCHESTRA}}.$$

In this paper, learning mainly focuses on the two following complementary dimensions: Since the delegation parameters $\Phi_t = (I_t, C_t, T_t, M_t)$ are explicitly available, learning can focus on two complementary dimensions: (i) **Task orchestration**, which determines what to do, what context to use, and which tools to employ. (ii) **Model routing**, which selects $M_t$ (the model to call) to balance performance and cost. In the following, we detail these two learning paradigms.

**Supervised fine-tuning (SFT) for task orchestration.** Given expert orchestration trajectories $\{(s_t, a_t^\star)\}$, we fine-tune the Orchestrator by behavior cloning:

$$\theta^\star = \arg\max_\theta \sum_t \log p_\theta(a_t^\star \mid s_t),$$

where $a_t^\star$ is the expert action (either $\text{Delegate}(\Phi_t^\star)$ or $\text{Finish}(y^\star)$). In our setup, SFT primarily distills *task orchestration*: improving subtask decomposition and the synthesis of $(I_t, C_t, T_t)$, i.e., producing better working memory, and more appropriate tool subsets for each step. We would like to note that in this work, we prioritize showing the potential of training a specialized orchestrator, thus employing a straightforward SFT approach. Note that others can employ any training methods like GRPO (Shao et al., 2024) to improve the task orchestration capability.

**Iterative In-context Learning for Cost-aware Orchestration.** Beyond parameter updates, we also optimize orchestration *without* changing model weights by learning the Orchestrator's *instruction* (prompt) through iterative interaction. Concretely, we treat the Orchestrator instruction $I^{\text{main}}$ as the learnable object and run AORCHESTRA in the environment to collect trajectories $\tau = \{(s_t, a_t, o_t)\}_{t=0}^T$ together with outcome metrics, including task performance and execution cost. An optimization model then analyzes these trajectories and proposes prompt edits $\Delta I$ to update the instruction:

$$I_{k+1}^{\text{main}} = \text{OPTIMIZE}\big(I_k^{\text{main}}, \tau_k, \text{Perf}(\tau_k), \text{Cost}(\tau_k)\big),$$

where $k$ indexes optimization rounds. By repeatedly rolling out the updated Orchestrator in the environment for $N$ rounds, this process improves cost-aware orchestration behavior (e.g., model compiler/routing decisions and tool usage patterns) and aims to discover Pareto-efficient trade-offs between performance and cost.

## 4. Experiments

### 4.1. Experiment Setup

**Benchmarks.** We evaluate our method on three challenging agentic benchmarks that span diverse interactive settings: (1) **Terminal-Bench 2.0** (Team, 2025), which places agents in a Linux terminal with an interactive Bash shell, requiring them to execute command-line operations to complete multi-step real-world tasks; (2) **SWE-Bench-Verified** (Jimenez et al., 2023), which assesses software engineering on real GitHub projects, where agents must localize bugs, implement patches, and satisfy the provided test suites under realistic coding environment; and (3) **GAIA** (Mialon et al., 2023) validation set, a generalist benchmark that tests an agent's ability to solve real-world tasks requiring multi-step reasoning and tool use. We report pass@1 and pass@3 for

*Table 1.* Comparison between AORCHESTRA and baseline agentic systems on GAIA, Terminal-Bench 2.0, and SWE-Bench-Verified under various models. The best results are in **bold**.

| Methods | Model Setup | GAIA | | Terminal-Bench 2.0 | | SWE-Bench-Verified | | |
|---|---|---|---|---|---|---|---|---|
| | | Pass@1 | Pass@3 | Pass@1 | Pass@3 | Pass@1 | Pass@3 | Avg. Pass@1 |
| ReAct | Gemini-3-Flash | 49.09 | 66.06 | 28.57 | 47.14 | 64.00 | 82.00 | 47.22 |
| | DeepSeek-V3.2 | 46.70 | 71.51 | 20.00 | 32.86 | 48.00 | 87.00 | 38.23 |
| | Claude-4.5-haiku | 47.88 | 62.42 | 20.00 | 37.14 | 63.00 | 87.00 | 43.62 |
| OpenHands | Gemini-3-Flash | 66.06 | 72.73 | 31.43 | 51.43 | 48.00 | 66.00 | 48.49 |
| | DeepSeek-V3.2 | 63.64 | 72.12 | 21.43 | 35.71 | 60.00 | 75.00 | 48.35 |
| | Claude-4.5-haiku | 54.55 | 61.21 | 12.85 | 25.71 | 68.00 | 83.00 | 45.13 |
| Mini-SWE | Gemini-3-Flash | 58.18 | 68.48 | 34.29 | 50.00 | 56.00 | 85.00 | 49.49 |
| | DeepSeek-V3.2 | 50.30 | 63.63 | 30.00 | 48.57 | **84.00** | **89.00** | 54.76 |
| | Claude-4.5-haiku | 40.61 | 60.00 | 24.29 | 28.57 | 44.00 | 83.00 | 36.30 |
| Claude Code | Gemini-3-Flash | – | – | 32.86 | 48.57 | 22.00 | 42.00 | 27.43 |
| | Claude-4.5-haiku | – | – | 34.29 | 45.71 | 25.00 | 41.00 | 29.65 |
| AORCHESTRA | Gemini-3-Flash | **80.00** | **86.06** | **52.86** | **57.14** | 82.00 | 86.00 | **71.62** |
| | DeepSeek-V3.2 | 67.87 | 80.00 | 31.43 | 42.86 | 76.00 | 82.00 | 58.43 |
| | Claude-4.5-haiku | 60.61 | 73.90 | 35.71 | 45.71 | 70.00 | 84.00 | 55.44 |

all benchmarks respectively. We report more details about how we use these datasets for evaluation in Appendix A.1. We also detail the tools we used for each benchmark in Appendix D.

**Model & Baselines.** We compare our method against representative frameworks: (1) **ReAct** (Yao et al., 2022), a simple single-agent system directly build on ReAct that interleaves reasoning and actions; (2) **OpenHands** (Wang et al., 2024), a common-used open agent platform for solving diverse real-world tasks; (3) **mini-SWE-agent** (Yang et al., 2024), a minimalistic coding agent designed to solve GitHub issues and more; and (4) **Claude Code** (Anthropic, 2025), a production-grade agentic CLI that supports spawning pre-defined sub-agents for task decomposition and context isolation. For each agentic system, we employ the following frontier language models, including two strong models (`Gemini-3-Flash` and `DeepSeek-V3.2` (Liu et al., 2025a)) and a smaller model (`Claude-4.5-haiku`). We report the implementations of baselines in Appendix A.3.

**Implementation.** Across all experiments, we set $max\_attempt = 10$ for the orchestrator and $max\_step = 50$ for the subagent. We set $max\_step = 500$ for all baselines for a fair comparison. For the training-free setting, we detail our designs in Appendix B, which includes all prompts for the orchestrator and sub-agent we use across three benchmarks. Following prior works (Anthropic, 2025; Sun et al., 2025), the orchestrator would receive the summary results returned by each sub-agent.

For SFT training, we fine-tune Qwen3-8B (Yang et al., 2025) to improve its orchestration capability in non-thinking mode. We use TaskCraft (Shi et al., 2025a) as the seed

dataset and employ Gemini-3-Flash to collect 2K orchestration trajectories for SFT training. During SFT, we perform full-parameter fine-tuning under LLamaFactory framework (Zheng et al., 2024) for 2 epochs with a learning rate of 1e-5, with more details in Appendix A.2.

For in-context learning, we use `Claude Sonnet 4.5` as an optimization model to iteratively update the Orchestrator instruction. We run 5 optimization rounds; in each round, we collect 6 interaction trajectories for analysis. After each round, the prompt that achieves the best cost–performance trade-off (highest performance with lower cost) is selected for initializing the next round.

### 4.2. Main Results

Table 1 presents the main results of AORCHESTRA compared to baseline agentic systems on three benchmarks (GAIA, Terminal-Bench 2.0, and SWE-Bench-Verified), evaluated by pass@1/pass@3 metric. For AORCHESTRA, we use `Gemini-3-Flash` as the orchestrator and use only one model as subagent choices here for comparison. Overall, we find that AORCHESTRA consistently outperforms the baselines on all environments. AORCHESTRA outperforms the best baselines by an average of 22.13% pass@1 with `Gemini-3-Flash` across three benchmarks.

**GAIA Results** GAIA measures the ability of a general-purpose agent to solve real-world tasks, such as multi-hop searching, file processing, and multimodal operations. In such an environment, AORCHESTRA achieves the strongest performance against all baselines. Specifically, with `Gemini-3-Flash` as both the orchestrator and the

sub-agent model, AORCHESTRA achieves 80.00 pass@1 and 86.06 pass@3, which represents the best performance among all baselines. Under the same `Gemini-3-Flash` backbone, AORCHESTRA raises pass@1 by 13.94 points absolute over the strongest baseline framework, `OpenHands`, increasing the result from 66.06 to 80.00. Even with a less powerful model `Claude-4.5-haiku` as the subagent model, it still attains 60.61 pass@1 on GAIA, confirming that the observed improvements are not confined to the most capable model configuration. We do not evaluate Claude Code for GAIA because it is designed as a production-level coding agent, and thus its corresponding result is left blank. We present a case study on GAIA in Figure 4 and Appendix C.

**Terminal-Bench 2.0 Results**   Terminal-Bench assesses an agent's ability to operate in computer terminal environments inspired by real-world workflows. On this benchmark, AORCHESTRA with `Gemini-3-Flash` achieves 52.86 pass@1 and 57.14 pass@3. This is an absolute improvement of 18.57 points in pass@1 over the strongest baseline in Table 1, `Mini-SWE` with 34.29 pass@1. Beyond the `Gemini-3-Flash` setting, AORCHESTRA remains competitive under other backbones, with performance that is comparable to or better than specialized coding agentic systems such as `Claude Code`.

**SWE-Bench-Verified Results**   SWE-Bench-Verified evaluates an agent's ability to resolve real issues in open-source repositories by producing code patches that pass the provided tests. On this benchmark, AORCHESTRA achieves strong performance across backbones and is competitive with the best baseline systems. With `Gemini-3-Flash`, AORCHESTRA reaches 82.00 pass@1 and 86.00 pass@3, outperforming `ReAct` and `OpenHands` under the same model setting. Compared with `Mini-SWE`, which is designed for software tasks, AORCHESTRA remains competitive, and it consistently achieves over 70.00 pass@1 across all three model backbones.

## 4.3. Advantage Analysis of AORCHESTRA

In this section, we present analyses that demonstrate the benefits of AORCHESTRA for dynamically creating specialized sub-agents, particularly in terms of working memory and capabilities. Overall, our findings indicate that explicitly passing context from the orchestrator to sub-agents yields performance gains (Sec. 4.3.1). We also show that selecting different models for different tasks can achieve a cost-performance Pareto (Sec. 4.3.2), and that diverse implementations of sub-agents consistently contribute to overall improvement (Sec. 4.3.3)

*Table 2.* Context-control ablation for sub-agent invocation. We isolate the effect of context inheritance by only changing the `Context` field passed to sub-agents, while keeping the sub-agent model, tools, and system prompt identical across settings.

| Setting | Level 1 | Level 2 | Level 3 | Avg. |
|---|---|---|---|---|
| No-Context | 89.47 | 81.48 | 75.00 | 86.00 |
| Full-Context | 94.74 | 77.78 | 75.00 | 84.00 |
| **Ours** | **100.00** | **88.89** | **75.00** | **96.00** |

### 4.3.1. ADVANTAGE 1: CONTEXT SHARING

In AORCHESTRA, the orchestrator explicitly and dynamically passes curated context to each created sub-agent. To evaluate the effectiveness of this design, we compare it with two varients: **No-Context**, where each subagent only receives a task instruction, and **Full-Context**, where each subagent inherits all context from the main agent. Note that these two approaches are also commonly used in prior systems (Sun et al., 2025; Anthropic, 2025). Here, we conduct analysis on GAIA, and sample 50 samples from the validation set.

Table 2 indicates that it is necessary to regard context as an important component of subagent and abstract it into one of the four tuples. In particular, we find that No-Context fails due to the lack of critical execution traces and fine-grained cues from previous steps, whereas Full-Context often introduces irrelevant information and aggravates context degradation. In contrast, by allowing the orchestrator to select and compress only task-relevant history, our method provides a cleaner context and achieves the highest score.

### 4.3.2. ADVANTAGE 2: A LEARNABLE ORCHESTRATOR

*Table 3.* Main results. ReAct is evaluated with a single specified LM per run, while other systems may use either a single LM or a mixed-LM pool. ICL denotes context learning

| System | LM | Acc. | Avg. Cost |
|---|---|---|---|
| ReAct | Claude-4.5-sonnet | 53.93 | 0.190 |
| | Claude-4.5-haiku | 47.88 | 0.066 |
| | Gemini-3-Flash | 49.09 | 0.070 |
| | GPT-5-mini | 54.55 | 0.052 |
| | Deepseek-v3.2 | 46.70 | 0.027 |
| Ours | Gemini-3-Flash | **80.00** | 0.79 |
| | Claude-4.5-sonnet | 71.52 | 0.91 |
| | GPT-5-mini | 67.27 | 0.28 |
| | Deepseek-v3.2 | 67.87 | 0.14 |
| Ours (Gemini-3-Flash) | Mixed | 72.12 | 0.70 |
| Ours (ICL) | Mixed | **75.15** | **0.25** |
| Ours (Qwen3-8B) | Gemini-3-Flash | 56.97 | 0.36 |
| Ours (SFT) | Gemini-3-Flash | **68.48** | **0.68** |

**Supervised fine-tuning (SFT) for task orchestration**   A practical consideration of AORCHESTRA is that orchestration quality depends on the main agent's ability to

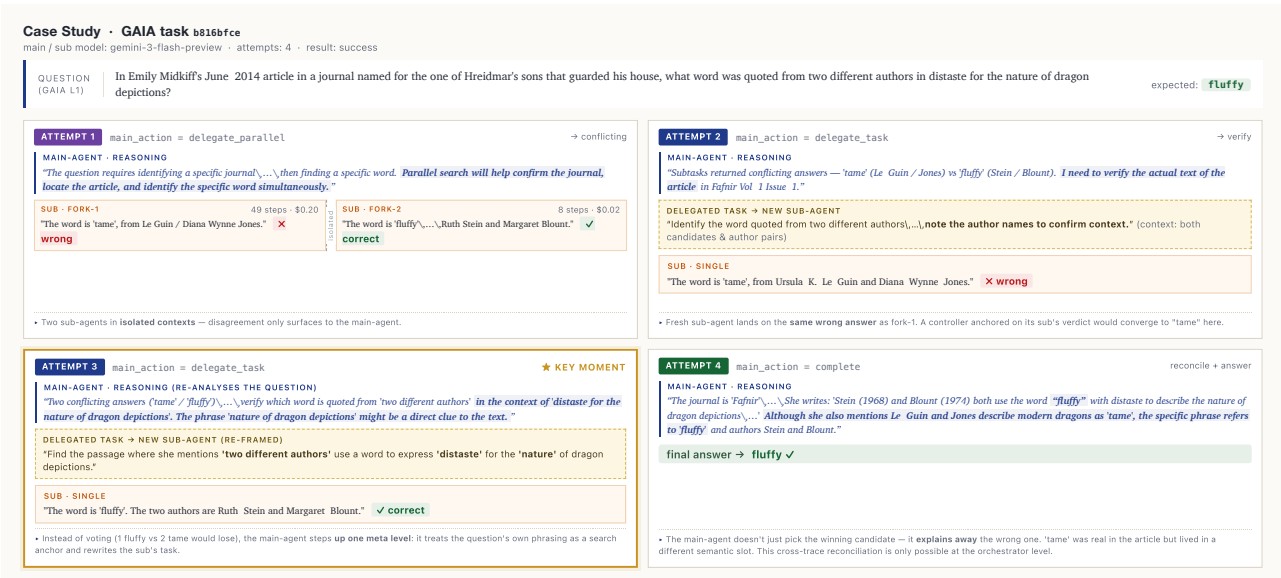

*Figure 4.* A trajectory excerpted from one of our GAIA case studies on task `b816bfce`.

decompose goals and synthesize high-quality delegation tuples. To probe this sensitivity, we replace the main agent with a weaker model, `Qwen3-8B`, while keeping `Gemini-3-Flash` as the sub-agent executor. As shown in Table 3, this setting (OURS (QWEN3-8B)) achieves only 56.97% accuracy at $0.36 average cost, substantially below using a strong main agent (OURS with `Gemini-3-Flash` reaches 80.00%). This gap suggests that weaker main agents struggle with task decomposition and delegation synthesis. We then fine-tune `Qwen3-8B` for orchestration via SFT, which yields a large improvement from 56.97% to 68.48% (Table 3). This gain indicates that orchestration is a learnable skill that can be efficiently improved, and further highlights the practicality of our design: the clean `Delegate` interface makes it straightforward to distill and enhance orchestration behavior.

**Behavior analysis of SFT.** We compare the behavior and trajectories of the orchestrator before and after SFT. Through a careful analysis, we find that the SFT gain mainly comes from a change in how the orchestrator aggregates the final answer, and manifests as two recurring behavioral shifts: (i) when the sub-agent does not return an explicit `finish_result`, the post-SFT orchestrator extracts the final answer directly from the partial trace instead of launching another retry; (ii) when later attempts disagree with the first one, the post-SFT orchestrator no longer anchors on the first-attempt answer but adopts the result repeatedly verified by subsequent attempts. We illustrate each with one example. For (i), on the USGS American-Alligator query, the pre-SFT orchestrator spent nine consecutive attempts that each exhausted the 50-step sub-agent budget without

producing an explicit `finish_result`, and finally committed to a wrong answer (1948); the post-SFT orchestrator instead reads the relevant snippet directly out of the search observation and emits the correct answer (1954) by the second attempt. For (ii), on a storage-cost question, the pre-SFT orchestrator obtained 0.03 in four later attempts yet ultimately regressed to the first-attempt answer (0.01), while the post-SFT orchestrator stably converges to 0.03 within three attempts.

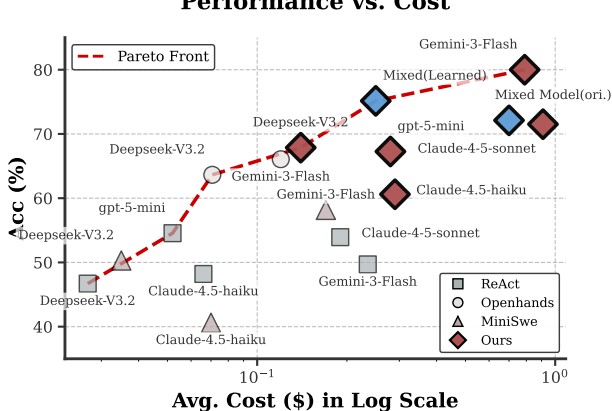

*Figure 5.* **Pareto front curve of GAIA.** We plot GAIA accuracy and average cost per task (USD, log scale). Each point corresponds to a configuration, and the dashed curve indicates the Pareto frontier formed by AORCHESTRA across different model routing choices.

**In-context Learning for Cost-aware Orchestration** Another advantage of AORCHESTRA is the ability to balance

cost and performance through step-wise model routing. Table 3 shows that using different sub-agent models leads to markedly different accuracy–cost profiles, making it important for the orchestrator to be sensitive to such trade-offs. We therefore apply a Pareto-oriented context learning procedure that iteratively optimizes the Orchestrator instruction from interaction trajectories with both performance and monetary cost feedback. The resulting policy improves AORCHESTRA while reducing cost: under the mixed-model setting, Ours (ICL) improves accuracy from 72.12% to 75.15% while lowering average cost from $0.70 to $0.25 (Table 3), demonstrating that simple prompt-level learning can jointly enhance performance and efficiency. At the system level, Figure 5 further shows that AORCHESTRA naturally yields strong Pareto-efficient operating points: across different model choices, our configurations form the Pareto frontier, indicating a systematically improved cost–performance trade-off over the baselines.

**Behavior analysis of ICL.** We compare the behavior and trajectories of the orchestrator before and after ICL. After a careful analysis, we find that the learned orchestrator exhibits two specific phenomena in its routing policy: (i) an explicit verification step is inserted before committing the final answer; (ii) before escalating to a stronger model, the orchestrator first diagnoses whether the failure was an execution issue or a sub-agent instruction issue—when it is the latter, it rewrites the sub-agent instructions under the same or a cheaper model and retries, escalating only when a true capability gap is identified. We illustrate each with one example. For (i), on tasks such as a BBC Earth bird-species query where the pre-ICL policy already converges but occasionally calls a more expensive model just to "double-check", the added verification step lets the orchestrator reach the same correct answer with fewer expensive calls. For (ii), on a previously failed quiz-image scoring task, the orchestrator no longer escalates the model directly but rewrites the sub-agent instructions and retries, recovering the correct answer without invoking the most expensive model.

### 4.3.3. ADVANTAGE 3: PLUG-AND-PLAY SUBAGENTS

Here, we aim to verify the framework-level pluggability of our approach. Specifically, we replace the execution backend of the sub-agent with different agent frameworks, such as ReAct-style and Mini-SWE-style with `Gemini-3-Flash` as the orchestrator on Terminal-Bench, following the setups in Appendix A.1.

Table 4 shows that when different sub-agent backends are used, AORCHESTRA maintains stable performance and consistently outperforms the corresponding baselines. This design allows sub-agents to be as pluggable modules, enabling the system to remain robust without depending on any particular sub-agent implementation.

*Table 4.* Evaluating plug-and-play sub-agents with a fixed orchestrator. We use Gemini-3-Flash to test the robustness and reusability of different sub-agent implementations.

| System | Easy | Medium | Hard | Acc |
|---|---|---|---|---|
| **Standalone baselines** | | | | |
| ReAct | 50.00 | 34.09 | 16.67 | 28.57 |
| Mini-SWE-Agent | 50.00 | 40.91 | 20.83 | 34.29 |
| Claude Code | 50.00 | 41.86 | 16.67 | 32.86 |
| **Orchestrator with plug-in sub-agents** | | | | |
| ReAct-style SubAgent | 50.00 | **63.63** | 20.83 | **48.57** |
| Mini-SWE-style SubAgent | **100.00** | 47.73 | **33.33** | 44.29 |

## 5. Conclusion

We present AORCHESTRA, an orchestration-centric agentic system that automates sub-agent creation via a unified four-tuple interface (Instruction, Context, Tools, Model). Treating sub-agents as dynamically creatable units lets the orchestrator spawn task-tailored executors on demand with specialized memory and capabilities, enabling task-sufficient context, plug-and-play sub-agents, and a learnable orchestrator optimized via supervised fine-tuning and in-context learning. Empirically, AORCHESTRA delivers consistent gains across GAIA, Terminal-Bench, and SWE-Bench-Verified, outperforming established baselines by an average of 16.28% pass@1 with Gemini-3-Flash, validating orchestration-centric design for complex, long-horizon tasks.

## Impact Statement

This paper presents work whose goal is to advance the field of machine learning by making agentic systems more modular, controllable, and efficient through a unified orchestration abstraction. By modeling sub-agents as a four-tuple interface and enabling on-demand creation of specialized executors with curated context, our framework supports systematic studies of long-horizon tool use, cross-framework plug-and-play robustness, and cost–performance trade-offs across heterogeneous environments. Such orchestration capabilities could be misused to automate harmful workflows or to carry out actions that produce harmful benefits for certain actors. To reduce these risks, our design emphasizes explicit, auditable interfaces for delegation and tool invocation, along with curated context and constrained tool access, which together encourage more transparent, controllable, and accountable agent behavior.

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

# A. Implemention Details

## A.1. Datasets

- **GAIA** (Mialon et al., 2023): GAIA benchmarks general AI assistants on realistic, tool-augmented questions (often involving web browsing and multi-step reasoning). We evaluate AORCHESTRA and compare with other baselines on the GAIA validation split, which contains a total of 165 tasks.

- **Terminal Bench 2.0** (Team, 2025): Terminal-Bench evaluates agents on end-to-end, real-world workflows in a sandboxed command-line environment, graded by executable tests. We evaluate AORCHESTRA and compare with other baselines on the Terminal-Bench2.0 test split, which contains a total of 89 tasks. In the main experiments, we randomly sample 70 tasks dut to cost reasons.

- **SWE-Bench-Verified:** (Jimenez et al., 2023) SWE-Bench Verified measures autonomous software engineering by asking agents to generate patches that resolve real GitHub issues in real repositories, verified by running tests; the Verified split is human-screened to remove problematic cases. We evaluate AORCHESTRA and compare with other baselines on the SWE-Bench verified version test split, which contains a total of 500 tasks. We randomly sample 100 tasks for evaluation due to cost reasons.

## A.2. SFT Hyper-parameters.

We use the following hyperparameters during the experiments in Table 5.

| Hyperparams | Values | Hyperparams | Values |
|---|---|---|---|
| learning rate | 1e-5 | weight decay | 0.05 |
| warmup ratio | 0.1 | max length | 16K |
| lr scheduler | cosine | batch size | 64 |
| epoch | 2 | BF16 | True |
| Deepspeed | zero3 | tool-call template | Hermes |

*Table 5.* SFT Hyperparameters used.

## A.3. Baseline Implementations

For baseline implementations, we evaluate a diverse set of widely-used agentic frameworks. During our experiment, we found that `Claude Code` is not well-suited for GAIA open-world multi-hop question answering due to its architecture and intended usage pattern. Therefore, we did not report the results of `Claude Code` in the main experiments.

In addition, we find that `DeepSeek-V3.2` exhibits poor native compatibility with CLAUDE CODE based on our initial investigation and empirical trials. Therefore, we exclude this experiment in Table 1.

For the Terminal-Bench and SWE-Bench evaluations, we leverage the `Harbor` scaffold to run `MiniSWE`, `OpenHands`, and `Claude Code` under a unified execution interface.

# B. Prompts

## B.1. Main Agent Prompts

### B.1.1. GAIA MAIN AGENT PROMPT

---

**GAIA Main Agent Prompt**

---
```
"""
[GAIA BENCHMARK - QUESTION ANSWERING TASK]
You are the MainAgent (Orchestrator). Your task is to
solve the given QUESTION by decomposing it into subtasks and delegating each to a sub-agent.

DECISION PROCESS:
1. REVIEW the SUBTASK HISTORY below - check status, result, and key findings of each attempt
2. EVALUATE: Do the results SUFFICIENTLY answer the QUESTION?
   - If any subtask returned a valid result with status "done" → Consider using 'complete'
```

```
      - If subtask status is "incomplete" → Review its key findings to see what was accomplished
3. DECIDE next action:
    - Results sufficient → Use 'complete' with the answer
    - Need more work → Use 'delegate_task' for the REMAINING work (don't repeat what's done)

BUDGET AWARENESS:
- You have LIMITED attempts (see Progress below)
- Each delegation costs time and resources - choose models wisely based on task complexity
- If a result looks correct and was verified, trust it and complete

==== MODEL SELECTION GUIDE ====
{model_pricing_table}

Model Selection Strategy:
- Choose cheaper models for simple tasks
- Choose more capable models for complex reasoning or critical attempts

==== Progress ====
[Attempt {attempt_index}/{max_attempts}] Remaining {remaining_attempts} attempts
Budget is limited. Make each attempt count.

==== QUESTION ====
{instruction}

==== SUBTASK HISTORY ====
{subtask_history if subtask_history else "No subtasks completed yet."}

==== AVAILABLE TOOLS ====
{tools_description}

==== OUTPUT ====
ANSWER FORMAT: GAIA requires precise, concise
answers (single word, number, or short phrase). Do NOT
include explanations in the answer field.

Return JSON:

If results are SUFFICIENT:
{{
  "action": "complete",
  "reasoning": "The subtask results show [X], which answers the question",
  "params": {{ "answer": "concise answer" }}
}}

If more work is NEEDED:
{{
  "action": "delegate_task",
  "reasoning": "We have [X] from previous attempts, but still need [Y] to answer the question",
  "params": {{
    "task_instruction": "A SPECIFIC, ACTIONABLE subtask
    (e.g., 'Extract second word from abstract of paper 2211.xxxxx')",
    "context": "Relevant findings from previous attempts",
    "model": "one of {sub_models}",
    "tools": ["GoogleSearchAction", "ExtractUrlContentAction",
    "ExecuteCodeAction", "ImageAnalysisAction", "ParseAudioAction"]
  }}
}}
}}
"""
```

## B.1.2. TERMINAL-BENCH MAIN AGENT PROMPT

**Terminal-Bench Main Agent Prompt**

```
"""
You are the MainAgent (Orchestrator). Your task is to
complete the given software installation/configuration task by delegating to SubAgents.

CRITICAL: CONTAINER LIFECYCLE
- Each SubAgent runs in a FRESH container - if you delegate_task again, the previous work will be lost
- When SubAgent reports status="done", use 'submit' immediately to run tests in that container

==== DECISION PROCESS ====
1. READ the original TASK carefully - identify ALL requirements and edge cases
2. REVIEW SUBTASK HISTORY - check status and completed steps
3. VERIFY SubAgent's work against TASK requirements:
    - Did SubAgent test ALL requirements mentioned in TASK?
```

```
        - Did SubAgent test edge cases? (e.g., if TASK mentions "keyboard interrupt", was it actually tested?)
        - Are SubAgent's "completed" items actually addressing the TASK requirements?
   4. DECIDE:
        - status="done" AND verification passes → Use 'submit'
        - status="done" BUT verification passes but some requirements are not met → Use 'delegate_task' to fix
        - status="partial" → Use 'delegate_task' with context about what worked/failed

   {budget_warning}

   ==== MODEL SELECTION ====
   {model_pricing_table}

   ==== Progress ====
   [Attempt {attempt_index}/{max_attempts}] Remaining {remaining_attempts} attempts

   ==== TASK ====
   {instruction}

   ==== SUBTASK HISTORY ====
   {subtask_history if subtask_history else "No subtasks completed yet."}

   ==== AVAILABLE TOOLS ====
   {tools_description}

   ==== OUTPUT ====
   Return JSON:

   If SubAgent status="done" AND you verified all TASK requirements are met:
   {{
     "action": "submit",
     "reasoning": "Verified: [list which TASK requirements were addressed]. Submitting.",
     "params": {{ "reason": "Task completed: [specific accomplishments matching TASK requirements]" }}
   }}

   If SubAgent status="done" BUT verification shows gaps:
   {{
     "action": "delegate_task",
     "reasoning": "SubAgent claimed done but [specific gap]: TASK requires [X] but SubAgent only tested [Y]",
     "params": {{
       "task_instruction": "CRITICAL: Previous attempt missed [specific requirement].
                         You MUST: [exact steps to fix]",
       "context": " PREVIOUS SUBAGENT CLAIMED DONE BUT MISSED: [specific gap]\\n-
                   WORKED: [steps to keep]\\n-
                   MUST FIX: [what was missed]",
       "model": "one of {sub_models}"
     }}
   }}

   If SubAgent status="partial":
   {{
     "action": "delegate_task",
     "reasoning": "SubAgent made partial progress, need to continue with [remaining work]",
     "params": {{
       "task_instruction": "Continue from where previous SubAgent left off: [specific next steps]",
       "context": "From SUBTASK HISTORY:\\n-
                   WORKED: [steps to REPEAT]\\n-
                   FAILED: [approaches to AVOID]",
       "model": "one of {sub_models}"
     }}
   }}
   """
```

### B.1.3. SWE-BENCH MAIN AGENT PROMPT

**SWE-Bench Main Agent Prompt**

```
"""You are the MainAgent (Orchestrator) for a SWE-bench task.
Your goal is to fix a GitHub issue by delegating work to SubAgents.

==== TASK ====
{instruction}

REPOSITORY: {repo}
INSTANCE: {instance_id}

==== DECISION PROCESS ====
```

```
   1. READ the TASK carefully - understand the GitHub issue and what needs to be fixed
   2. REVIEW SUBTASK HISTORY - check SubAgent's progress, completed steps, and test results
   3. VERIFY against TASK requirements:
      - Did SubAgent locate the buggy code?
      - Did SubAgent make appropriate code changes?
      - Did SubAgent run tests and confirm the fix works?
   4. DECIDE:
      - status="done" AND tests pass → Use 'submit'
      - status="done" BUT tests fail or incomplete → Use 'delegate_task' to fix remaining issues
      - status="partial" → Use 'delegate_task' with guidance on next steps

   CRITICAL: SWE-BENCH CONTAINER BEHAVIOR
   - When SubAgent reports status="done" with passing tests, use 'submit' to trigger final evaluation
   - 'submit' runs the official test suite (FAIL_TO_PASS + PASS_TO_PASS tests) to determine success

   ==== MODEL SELECTION ====
   {model_pricing_table}

   ==== Progress ====
   [Attempt {attempt_index}/{max_attempts}] Remaining {remaining_attempts} attempts
   {budget_warning}

   ==== SUBTASK HISTORY ====
   {subtask_history if subtask_history else "No subtasks completed yet."}

   ==== AVAILABLE TOOLS ====
   {tools_description}

   ==== OUTPUT ====
   Return JSON:

   If SubAgent status="done" AND tests pass:
   {{
     "action": "submit",
     "reasoning": "Verified: [what was fixed, which tests passed]. Submitting for evaluation.",
     "params": {{ "reason": "Fix verified: [specific fix description]" }}
   }}

   If SubAgent status="done" BUT tests fail or incomplete:
   {{
     "action": "delegate_task",
     "reasoning": "SubAgent reported done but [specific issue]: tests show [failure details]",
     "params": {{
       "task_instruction": "CRITICAL: Previous fix incomplete. [specific next steps needed]",
       "context": " ISSUE: [what failed]\\n-  DONE: [completed work]\\n-  TODO: [remaining work]",
       "model": "one of {sub_models}"
     }}
   }}

   If SubAgent status="partial":
   {{
     "action": "delegate_task",
     "reasoning": "SubAgent made partial progress: [summary]. Need to [next steps]",
     "params": {{
       "task_instruction": "Continue: [specific next steps based on SUBTASK HISTORY]",
       "context": "From previous attempt:\\n-  WORKED: [keep these]\\n-  FAILED: [avoid these]",
       "model": "one of {sub_models}"
     }}
   }}
}}
"""
```

## B.2. Sub-Agent Prompts

### B.2.1. GAIA SUB-AGENT PROMPT

**Gaia Sub-Agent Prompt**

```
ORCHESTRA_GAIA_PROMPT = """
==== Progress ====
[Step {current_step}/{max_steps}] Remaining {remaining_steps} steps
{budget_warning}

==== Your Task (from MainAgent) ====
{task_instruction}
```

```
==== Context ====
{context}

==== Original Question (for reference) ====
{original_question}

==== Available Tools ====
{action_space}

==== Guidelines ====
1. Focus on completing YOUR TASK above
2. Think step by step before outputting an action
3. Write key observations to the "memory" field
4. Use print() in ExecuteCodeAction to see computation results
5. Once done, use 'finish' IMMEDIATELY

BUDGET: When remaining_steps <= 5, use 'finish' NOW!

==== Output Format ====
```json
{{
    "action": "<tool_name>",
    "params": {{}},
    "memory": "<observations>"
}}
```

==== Memory ====
{memory}

==== Current Observation ====
{obs}
"""
```

## B.2.2. TERMINAL-BENCH SUB-AGENT PROMPT

**Terminal-Bench Sub-Agent Prompt**

```
"""
==== Progress ====
[Step {current_step}/{max_steps}] Remaining: {remaining_steps} step(s)
{budget_warning}
If you run out of steps without "finish", your work is lost and marked as timeout.

==== Your Task (from MainAgent) ====
{task_instruction}

==== Context (from previous attempts) ====
{context}
Use this info: repeat what WORKED, avoid what FAILED.

==== Original Question (for reference) ====
{original_question}

==== Action Space ====
{action_space}

==== Memory ====
Recent memory:
{memory}

==== Current Observation ====
{obs}

==== Thinking ====
Think step by step before outputting an action. Write key reasoning in memory for future steps.

==== Action Guidelines ====
You have TWO actions available:

1. **execute** - Run shell commands and observe results
   - Use this to install packages, configure services, verify status, etc.
   - Example: "apt update && apt install -y nginx"

2. **finish** - Report your progress to MainAgent
   - Use when task is COMPLETE (status="done")
```

```
    - Use when you made PROGRESS but need more work (status="partial")
    - MUST use before running out of steps! Your work is LOST if you timeout.

**What to report in finish:**
- completed: List SUCCESSFUL steps that WORKED (e.g., ["apt update succeeded", "nginx installed"])
- issues: List FAILED attempts with WHY (e.g., ["nginx -v failed: command not found"])
- message: Brief summary of current state

This info helps the NEXT SubAgent know what to repeat and what to avoid.

==== Output Format ====
CRITICAL: You MUST reply with ONLY a JSON object. No explanations, no markdown, no other text.

For execute:
{{"action": "execute", "params": {{"command": "your shell command"}}, "memory": "key findings"}}

For finish:
{{"action": "finish", "params": {{"status": "done|partial", "completed": [...],

"issues": [...], "message": "..."}}, "memory": "final notes"}}

"""
```

### B.2.3. SWE-BENCH SUB-AGENT PROMPT

**SWE-Bench Sub-Agent Prompt**

```
SWEBENCH_SUBAGENT_PROMPT = """
You are an autonomous software engineering agent tasked with solving GitHub issues.
You have access to a specialized command interface (ACI) for navigating, viewing, editing, and testing code.
You will work in a Docker container with the repository already cloned and checked out to the correct commit.

==== Progress ====
[Step {current_step}/{max_steps}] Remaining: {remaining_steps} step(s)
{budget_warning}
If you run out of steps without "finish", your work is lost and marked as timeout.

==== Your Task (from MainAgent) ====
{task_instruction}

==== Context (from previous attempts) ====
{context}

==== Current State ====
{state_info}

==== Command Reference ====
{command_docs}

=== FINISH (Report to MainAgent) ===
finish <status> <message>
    Report your progress back to MainAgent. Status MUST be one of:
    - done: Task completed successfully, tests pass
    - partial: Made progress but not finished (e.g., found bug but fix not working)

==== Memory ====
Recent memory:
{memory}

==== Current Observation ====
{observation}

==== OUTPUT FORMAT (STRICT) ====
You MUST output EXACTLY two sections in this order. No other text allowed.

DISCUSSION
<your reasoning here>

COMMAND
<single command here>

RULES:
- DISCUSSION must contain your step-by-step reasoning
- COMMAND must contain exactly ONE command on a single line
- After COMMAND line, do NOT add any explanation, examples, or comments
```

```
- Do NOT output anything after the command
"""
```

## B.2.4. SUB-AGENT SUMMARY PROMPT

**Sub-Agent Summary Prompt**

```
"""You are a trajectory summarizer. Review the SubAgent's execution trace.
Compare the execution trace against the original task requirements.

== ORIGINAL TASK ==
{original_question}

== EXECUTION TRACE ==
{trace_text}

== OUTPUT ==
Based on the trace, answer:
1. COMPLETED: What requirements from the original task were actually done?
2. REMAINING: What requirements are still missing or not properly tested?

Summarize in 5-10 bullets: key progress, problems, remaining issues.
Output ONLY bullets.Be specific and concise. Output ONLY the two sections above."""
```

## B.3. Learning Prompt

### B.3.1. STRATEGY OPTIMIZE PROMPT

**STRATEGY OPTIMIZE PROMPT**

```
STRATEGY_OPTIMIZE_PROMPT = """
You are optimizing the MainAgent strategy block for GAIA tasks.
Focus on selecting sub-models, deciding when to delegate, and managing cost vs performance.
You need to analysis the model ability and cost on this task,
and creat a better strategy for main agent to select cheaper model while keep the performance.

Current strategy block:
{strategy}

Evaluation summary:
- pass_rate: {pass_rate}
- avg_reward: {avg_reward}
- total_cost: {total_cost}

Recent trajectories (summarized):
{trajectories}

Write one improved strategy block ONLY. Output in XML:
<prompt>...strategy text...</prompt>
"""
```

### B.3.2. STRATEGY SELECT PROMPT

**STRATEGY SELECT PROMPT**

```
STRATEGY_SELECT_PROMPT = """
You are comparing two MainAgent strategy prompts for GAIA tasks.
Summarize each trajectory's strengths/weaknesses,
relate them to the strategy text,
then decide which strategy is better overall considering BOTH performance and cost.

# A (parent/best-so-far)
pass_rate: {pass_rate_a} / avg_reward: {avg_reward_a} / total_cost: {total_cost_a}
strategy: {strategy_a}
```

```
    trajectories: {traj_a}

    # B (new candidate)
    pass_rate: {pass_rate_b} / avg_reward: {avg_reward_b} / total_cost: {total_cost_b}
    strategy: {strategy_b}
    trajectories: {traj_b}

    Respond with XML:
    <analysis>your reasoning</analysis>
    <choose>A/B</choose>
    """
```

# C. Case Study

## C.1. GAIA Case Study

We walk through a single GAIA trajectory on a spreadsheet-analytics task (file `076c8171`) to illustrate how the orchestrator detects and recovers from sub-agent under-delivery: when the first sub-agent signals completion without returning the required value, the main-agent sharpens the instruction and re-delegates, succeeding on the second attempt.

### Case Study 1: Spreadsheet Analytics (Excel)

Find the **vendor type** with the **lowest revenue-to-rent ratio** from an Excel sheet of vendors in Liminal Springs mall.

#### Attempt 1 — Turn 1: Main Agent

```
thinking: I need to read the Excel file, find the vendor with the lowest revenue-to-rent ratio,
        and report the value in the 'Type' column.
        This is a data analysis task and requires using pandas to process the Excel file.

action: delegate_task
params: {
    "task_instruction": "1. Load the Excel file '076c8171-9b3b-49b9-a477-244d2a532826.xlsx' using pandas.
                         2. Calculate the ratio of monthly revenue to rent for each vendor.
                         3. Identify the vendor with the lowest ratio of revenue to rent.
                         4. Return the value in the 'type' column for that vendor.",
    "context": "The file contains vendors in Liminal Springs mall with their monthly revenue and rent.",
    "model": "model_1",
    "tools": ["ExecuteCodeAction"]
}
```

#### Attempt 1 — Turn 2: Subagent

```
exploration:
  Step 1: Execute code to load the Excel file
        -> Detect a formatting issue: the first row is "Zone 1" rather than the true header.

  Step 2: Adjust the reading method using header=1
        -> Successfully identify columns: 'Name', 'Type', 'Revenue', 'Rent', 'Opened'
        -> Preview shows vendors such as Rainforest Bistro, Panorama Outfitters, etc.

  Step 3: Subagent calls finish (but does not return the requested final value)
```

#### Attempt 1 — Turn 3: Summary

```
  Completed: Successfully loaded the Excel file and recovered the correct header structure (requires header=1),
  ↪  confirming columns Name / Type / Revenue / Rent / Opened.

  Not completed: Did not compute the revenue/rent ratio and did not return the Type for the minimum-ratio
  ↪  vendor.

  Error pattern: The subagent signaled completion (via finish while the key deliverable was missing, i.e., a
  ↪  mismatch between the completion signal and actual task completion.
\end{itemize}
```

#### Attempt 2 — Turn 1: Main Agent

```
thinking: Attempt 1 successfully loaded the file and identified the column names, but it did not
          compute the ratio or identify the minimum-ratio vendor.
          I need the sub-agent to execute the full pipeline and return the concrete result.

action: delegate_task
params: {
    "task_instruction": "1. Load the Excel file '076c8171-9b3b-49b9-a477-244d2a532826.xlsx' using pandas.
                          2. Calculate the 'revenue_to_rent_ratio' as Monthly Revenue divided by Monthly Rent.
                          3. Identify the vendor with the ABSOLUTE MINIMUM revenue-to-rent ratio.
                          4. Return the value in the 'Type' column for that SPECIFIC vendor.",
    "context": "Previous attempt confirmed the file contains columns: 'Name', 'Type', 'Revenue', 'Rent'.
                Ensure you use header=1 to correctly parse the file.",
    "model": "model_1",
    "tools": ["ExecuteCodeAction"]
}
```

## Attempt 2 — Turn 2: Subagent

```
exploration:
  Step 1: Load the Excel file and hit KeyError: 'Revenue'
          -> Realize we must re-load with header=1.

  Step 2-4: Debug the parsing / schema
          -> Confirm the correct column structure.

  Step 5: Run the full computation
          -> Compute Revenue/Rent for all vendors
          -> Sort to find the global minimum

  Step 6: finish with the concrete result
          Vendor: Carnivore Loan Specialists
          Type: Finance
          Revenue: 31000.0
          Rent: 50312.0
          Ratio: 0.616155
```

## Attempt 2 — Turn 3: Summary

```
  Completed: Correctly parsed the sheet (using header=1), computed ratios, and found the global minimum.

  Key result: The minimum-ratio vendor is \texttt{Carnivore Loan Specialists}; its Type is Finance (ratio =
  ↪   0.616155).

  Improvement source: The main agent explicitly required the absolute minimum and returning the Type for that
  ↪   specific vendor, reducing ambiguity.

\end{itemize}
```

## Attempt 3 — Turn 1: Main Agent

```
thinking: Attempt 2 identifies the minimum-ratio vendor as "Carnivore Loan Specialists",
          whose Type is "Finance". The result is clear and can be submitted directly.

action: complete
params: {
    "answer": "Finance"
}
```

## Attempt 3 — Turn 2: Review

```
The main agent directly reuses the concrete artifact from Attempt 2 and submits the answer without additional
↪   delegation.
```

**Final Answer:** Finance

# D. Tools And Action Space

This appendix details the action space of the AORCHESTRA framework, including the actions available to the Main Agent and Sub-Agents, as well as the tool inventory and execution constraints for each benchmark.

## D.1. Main Agent Action Space

The Main Agent is responsible for global task planning and subtask delegation. Its action space consists of two core actions.

**delegate_task.** This action delegates a well-scoped subtask to a specialized Sub-Agent. The parameter schema is defined as follows:

---
**Action Schema: delegate_task**

```json
{
  "type": "object",
  "properties": {
    "task_instruction": {
      "type": "string",
      "description": "Detailed, actionable instruction for the Sub-Agent"
    },
    "context": {
      "type": "string",
      "description": "Additional context distilled from prior attempts"
    },
    "model": {
      "type": "string",
      "description": "Model alias to use",
      "enum": ["model_1", "model_2", "..."]
    },
    "tools": {
      "type": "array",
      "items": { "type": "string" },
      "description": "Optional subset of tools exposed to the Sub-Agent"
    }
  },
  "required": ["task_instruction", "model"]
}
```
---

**complete.** This action submits the final answer and terminates the episode.

---
**Action: complete**

```json
{
  "action": "complete",
  "params": {
    "answer": "<final answer string>"
  }
}
```
---

## D.2. Sub-Agent Tools For each Benchmark

Table 6 summarizes the tools available to Sub-Agents in each benchmark and their corresponding constraints.

### D.2.1. GAIA TOOLS

In the GAIA benchmark, Sub-Agents are equipped with tools for web retrieval, code execution, and multimodal analysis.

- **GoogleSearchAction.** Performs web search via the Serper API.

  ---
  ```json
  {"action":"GoogleSearchAction",
   "params":{"query":"...","k":5,"gl":"us","hl":"en"}}
  ```
  ---

- **ExtractUrlContentAction.** Extracts webpage content via the Jina API.

*Table 6.* Tool inventory and constraints per benchmark.

| Benchmark | Tool Name | Constraints / Notes |
|---|---|---|
| GAIA | `GoogleSearchAction` | Serper API; max 5 results; 30s timeout |
| | `ExtractUrlContentAction` | Jina API; chunked for long pages; 50s timeout |
| | `ExecuteCodeAction` | Sandboxed in `workspace/temp`; 10s time-out |
| | `ImageAnalysisAction` | Vision LLM backend; supports URL & local files |
| | `ParseAudioAction` | Audio-capable LLM backend; multi-format support |
| | `finish` | Reports result/status to the Main Agent |
| Terminal-Bench | `execute` | Shell commands in Docker/E2B sandbox |
| | `finish` | Reports progress without triggering tests |
| SWE-Bench | `execute` | Shell commands in a Docker container |
| | `view_file` | Reads a file with line-range specification |
| | `edit_file` | File editing via string replacement |
| | `finish` | Reports progress to main agent |

```
{"action":"ExtractUrlContentAction",
 "params":{"url":"...","browse_query":"..."}}
```

- **ExecuteCodeAction.** Executes Python or Bash code in a sandboxed environment.

```
{"action":"ExecuteCodeAction",
 "params":{"code":"...","code_type":"python|bash","timeout_sec":10}}
```

- **ImageAnalysisAction.** Calls a vision-capable LLM backend to analyze images.

```
{"action":"ImageAnalysisAction",
 "params":{"query":"...","image_path":"..."}}
```

- **ParseAudioAction.** Calls an audio-capable LLM backend to process audio inputs.

```
{"action":"ParseAudioAction",
 "params":{"query":"...","audio_path":"..."}}
```

- **finish.** Reports subtask results back to the Main Agent.

```
{"action":"finish",
 "params":{"result":"...","status":"done|partial|blocked","summary":"..."}}
```

### D.2.2. TERMINAL-BENCH TOOLS

In Terminal-Bench, Sub-Agents execute shell commands inside Docker/E2B sandboxes using the following tools:

- **execute:** run shell commands and return outputs.

- **finish:** report intermediate progress without triggering tests.

### D.2.3. SWE-BENCH TOOLS

In SWE-Bench, Sub-Agents are equipped with code navigation and editing capabilities:

- **execute:** run shell commands (e.g., `git` operations and tests).

- **view_file:** read file content with a specified line range.

- **edit_file:** edit files via string replacement.

- **finish:** Report your progress back to MainAgent.

### D.3. Sandbox and Network Constraints

**Code Execution Sandbox.** `ExecuteCodeAction` executes code in an isolated directory (`workspace/temp`). Potentially destructive Bash operations are disallowed, including file deletion, privilege escalation, permission changes, root-level redirection, and system-level commands. The default execution timeout is 10 seconds.

**Network Constraints.** Web access is mediated exclusively through tool APIs. Web search is performed via the Serper API (`google.serper.dev`) with a 30-second timeout, while URL content extraction is handled via the Jina API (`r.jina.ai`) with a 50-second timeout. Raw HTTP requests are not directly exposed to agents.

**Terminal-Bench Sandbox.** Terminal-Bench supports Docker, E2B, and Daytona backends. The default execution timeout is 600 seconds, and the working directory is automatically inferred from the Dockerfile `WORKDIR` directive.

**SWE-Bench Sandbox.** Each SWE-Bench task runs in an isolated Docker container. The system automatically clones the target repository and checks out the specified base commit. Tests are executed with `pytest` under a 300-second timeout.

## E. Third-Party API and Model Pricing

### E.1. Third-Party APIs

We rely on a small set of third-party APIs to support web search and sandboxed agent environment creation/execution.

*Table 7.* Third-party APIs used in our system.

| API | Role | How it is used in AORCHESTRA |
|---|---|---|
| **Serper** (`serper.dev`) | Web search | Used as the primary search API for retrieving relevant webpages/snippets during GAIA-style information-seeking subtasks. |
| **Jina** (`jina.ai`) | Web content retrieval | Used for lightweight webpage fetching/reading (e.g., converting a URL into clean text for extraction) to support search-and-read subtasks. |
| **E2B** (`e2b.dev`) | Sandbox environment | Used to create isolated execution environments for agent tool use (e.g., running code or environment-dependent operations) with controlled resources. |

**Usage.** These APIs are invoked only through our tool interface; the main agent and sub-agents do not access external services directly.
**Reproducibility.** When applicable, we cache retrieved web content and log API responses/metadata (e.g., timestamps, query strings, and URLs) to ensure consistent evaluation.

### E.2. Model Pricing Table and Cost Accounting

**Learned Agent Instruction**

```
PRICES = {
    # openai: https://platform.openai.com/docs/pricing
    # anthropic: https://docs.anthropic.com/en/docs/about-claude/pricing
    # openrouter: https://openrouter.ai/
    "gpt-4o-mini": {"input": 0.00015, "output": 0.0006},
    "o3": {"input": 0.002, "output": 0.008},
    "o3-mini": {"input": 0.0011, "output": 0.0044},
    "gpt-5": {"input": 0.00125, "output": 0.01},
    "gpt-5-mini": {"input":0.00025, "output": 0.002},
    "claude-sonnet-4-20250514": {"input": 0.003, "output": 0.015},
    "moonshotai/kimi-k2": {"input": 0.000296, "output": 0.001185},
    "deepseek/deepseek-chat-v3.1": {"input":0.00025 , "output":0.001},
    "deepseek-chat": {"input":0.00025 , "output":0.001},
    "z-ai/glm-4.5": {"input": 0.00033, "output": 0.00132},
    "gemini-2.5-pro": {"input": 0.00125, "output": 0.01},
    "claude-4-sonnet": {"input": 0.003, "output": 0.015},
```

```
    "claude-sonnet-4-5": {"input": 0.003, "output": 0.015},
    "claude-4-5-haiku": {"input": 0.00088, "output": 0.0044},
    "claude-4-sonnet-20250514": {"input": 0.003, "output": 0.015},
    "gemini-2.5-flash": {"input": 0.0003, "output": 0.00252},
    "gemini-3-flash-preview": {"input": 0.0005, "output": 0.003},
    "gemini-3-pro-preview": {"input": 0.002, "output": 0.004},
    "gemini-2.5-flash-image": {"input": 0.0003, "output": 0.03},
}
```

