# OpenReview forum: "AOrchestra: Automating Sub-Agent Creation for Agentic Orchestration"
_ICML.cc/2026/Conference — ICML 2026 regular_

### Official Review · Reviewer_kQr7 · 2026-03-11

**Soundness:** 2
**Presentation:** 3
**Significance:** 2
**Originality:** 2
**Overall Recommendation:** 3
**Confidence:** 3

**Summary:**

This paper proposes an orchestration-centric agent framework for long-horizon tasks. The core idea is to abstract any agent into the following tuple: (Instruction, Context, Tools, Model).

The paper proposes to treat sub-agents not as isolated threads, but as dynamically instantiated executors created on demand by a central orchestrator.

**Compliance With Llm Reviewing Policy:**

Affirmed.

**Final Justification:**

Thank you for the detailed rebuttal. The authors claim that dynamically generating (I, C, T, M) is their main contribution. However, this does not reflect a practical agent setup, nor is it empirically well motivated in practice. Works on calling a large number of APIs (tools) and selecting appropriate ones dates back to 2023 (https://arxiv.org/abs/2305.15334), but important tools that get widely adopted in practice remain small. This work lacks a clear motivation for the extra complexity introduced, as the authors note in their rebuttal. Therefore, I'd like to maintain my rating.

**Key Questions For Authors:**

Please see above.

**Limitations:**

Yes.

**Strengths And Weaknesses:**

Strengths:

1. The proposed abstraction is clean and generalizable.

Weakness:

1. While the design is clean, the abstraction naturally exists in many agent systems already, which leaves the question of 1). How are these abstractions different from existing protocols? 2) If there are agent systems that follow different abstraction paradigms, what's the advantage of this type of orchestration?

2. 4.2 seems to show the power of this abstraction through training. However, most of the texts are used to describe the abstraction, which does not leave enough space for possible ablation and analysis on the design space, significantly reducing the technical depth of the paper.

3. Continuing with the comment above, the writing needs significant work as the current framing spend most of the main texts on definitions, abstractions, designs that are already widely adopted in the field, and the experiment analyses do not dive into technical design choices.

---

> ### Author Rebuttal · Authors · 2026-03-31
>
> We thank the reviewer for the constructive feedback. All supplementary tables referenced below are available at https://anonymous.4open.science/r/table-ACE8/.
>
> # Distinction from existing protocols and advantage over different paradigms.
> We provide a detailed mechanism-level comparison in our response to Reviewer td36, and summarize the core arguments here. We identify a fundamental distinction between a *routing* problem (dispatching an existing agent) and a *synthesis* problem (constructing a new agent on the fly). AutoGen, LangGraph, and CrewAI all support dynamic routing at runtime — selecting which agent handles the current task. However, the agents themselves are pre-defined at development time: AutoGen's agents are instantiated before being passed to GroupChat [R1]; LangGraph's graph nodes are defined before compile() [R2]; CrewAI's agent pool is fixed at Crew creation time [R3]. None of them support dynamically generating an agent's instruction, context, tool set, or model configuration based on the current task. AOrchestra generates the full sub-agent specification (I, C, T, M) from scratch at each delegation step.
>
> Because delegation is parameterized as an explicit structured tuple, the orchestration policy becomes a learnable object — SFT yields +10.9% on GAIA [T3], and ICL optimization improves accuracy by +3.0% while reducing cost by 18.5%. In systems where configurations are hard-coded, no such structured decision space exists to optimize over.
>
> Regarding the advantage over different abstraction paradigms: AOrchestra does not require the underlying system to follow any specific paradigm — this is the structural advantage of an "orchestration layer" over a "complete framework." Whether the backend is ReAct-style, SWE-Agent-style, or Claude Code-style, AOrchestra can plug it in as a sub-agent executor and deliver consistent gains (+8.6% to +24.3% [T1]) without any modification. It does not replace existing execution logic, but complements it with dynamic sub-agent synthesis at the orchestration level.
>
> # Technical depth and writing.
> We first clarify a possible misunderstanding: **all experiments in Section 4.2 (Main Results) are training-free** — the orchestrator undergoes no learning or optimization, and performance gains come entirely from the orchestration abstraction itself. SFT and ICL experiments appear in subsequent sections, demonstrating the potential of the orchestration policy as a learnable object. The two are presented progressively, not conflated.
>
> We agree with the reviewer's suggestion on ablation depth and space allocation. We have supplemented the factorized ablation analysis the reviewer expects:
>
> | Setting | I | C | T | M | Acc. | Δ vs Full |
> |---|---|---|---|---|---:|---:|
> | Baseline (no orchestration) | - | - | - | fixed | 60.0% | -24.0 |
> | w/o context curation | dynamic | raw full-context | dynamic | fixed | 84.0% | 0.0 |
> | w/o tool selection | dynamic | dynamic curated | all tools | fixed | 80.0% | -4.0 |
> | Full (I, C, T) | dynamic | dynamic curated | dynamic  | fixed | 92.0% | +8.0 |
> | Full (I, C, T, M) | dynamic | dynamic curated | dynamic | dynamic | 84.0% | — |
>
> Table 7: Factorized ablation on GAIA (50-task subset) with Gemini-3-Flash-Preview.
>
> Regarding "abstractions already widely adopted": we believe AOrchestra differs from existing frameworks in important ways. First, existing frameworks lack a dynamic Model routing mechanism to optimize overall cost — our ICL experiment demonstrates that dynamic model selection improves accuracy while reducing cost by 18.5%. Second, and more fundamentally, AOrchestra is not designed as an agent framework but as a framework-agnostic, plug-and-play orchestration layer that augments existing agent systems (+8.6% to +24.3% [T1]). Our four-tuple definition is a higher-level abstraction over existing agent orchestration — the goal is not to replace these frameworks, but to provide them with a unified, learnable orchestration interface.
>
> If accepted, we commit to compressing the abstraction definition sections in the camera-ready version and reallocating the space to the above analyses to strengthen technical depth.
>
> **[T1] Rebuttal Table 1— all available at https://anonymous.4open.science/r/table-ACE8/**

---

> > ### Author Rebuttal · Reviewer_kQr7 · 2026-04-03
> >
> > Thank you for the detailed rebuttal. The authors claim that dynamically generating (I, C, T, M) is their main contribution. However, this does not reflect a practical agent setup, nor is it empirically well motivated in practice. Works on calling a large number of APIs (tools) and selecting appropriate ones dates back to 2023 (https://arxiv.org/abs/2305.15334), but important tools that get widely adopted in practice remain small. This work lacks a clear motivation for the extra complexity introduced, as the authors note in their rebuttal. Therefore, I'd like to maintain my rating.

---

> > > ### Author Response · Authors · 2026-04-08
> > >
> > > Thank you for engaging with our rebuttal.Our contribution is not about selecting from a large pool of tools, which as the reviewer correctly notes is well-studied since Gorilla (2023). Our contribution is the joint, dynamic synthesis of (Instruction, Context, Tools, Model) as an integrated sub-agent specification at each delegation step. This is fundamentally different from tool selection in isolation.
> > >
> > > **1. Tool selection alone is not the bottleneck.** Our ablation shows that the majority of gains come from the combination of components, not any single one.
> > > **2.The practical motivation is empirically validated, not assumed.** AOrchestra delivers consistent gains when plugging in unmodified ReAct/Mini-SWE/Claude Code backends.
> > >
> > > We would be grateful if the reviewer could share which specific aspect of the practical motivation remains unconvincing, so that we may provide further evidence during the remaining discussion period.

---

### Official Review · Reviewer_td36 · 2026-03-12

**Soundness:** 2
**Presentation:** 3
**Significance:** 2
**Originality:** 2
**Overall Recommendation:** 3
**Confidence:** 4

**Summary:**

This paper studies orchestration in multi-agent LLM systems. AOrchestra is proposed as a framework where each agent is defined by a tuple consisting of Instruction, Context, Tools, and Model. A central orchestrator decomposes a task into subtasks and dynamically creates specialized sub-agents by configuring these elements. Each sub-agent then executes its assigned subtask using its own context and tools. This paper argues that predefined agent roles are often inflexible, and that dynamically constructing agents for subtasks can improve adaptability. The framework is evaluated on several agent benchmarks, including GAIA, SWE-Bench-Verified, and Terminal-Bench. This paper also explores learning the orchestration policy using supervised fine-tuning and cost-aware routing.

**Compliance With Llm Reviewing Policy:**

Affirmed.

**Final Justification:**

The authors have addressed most of my concerns with more complete empirical results. However, I still feel that the proposed method is not very fundamental or of strong long-term significance. In particular, at the current stage of the field, given the recent progress in advanced agentic systems, I am not fully convinced that this work can be effectively applied to real-world scenarios, especially considering the additional complexity it introduces.

**Key Questions For Authors:**

After SFT of the orchestrator, what changes are observed in the orchestration behavior (e.g., number of agents spawned, decomposition strategy)?

**Limitations:**

No. The paper could more explicitly discuss limitations of dynamic multi-agent orchestration, such as increased orchestration overhead, coordination failures, or redundant exploration by multiple agents.

**Strengths And Weaknesses:**

### Strengths:
1. In general, the paper is clearly written and easy to follow.
2. This paper represents an agent as a tuple (Instruction, Context, Tools, Model). This abstraction provides a simple formulation of how task-specific agents are constructed and separates orchestration from execution.
3. As LLM-based agents are increasingly used for long-horizon tasks that involve tools and multiple reasoning steps, orchestration of multiple agents is an important systems proble.

### Weaknesses:
1. The main idea of dynamically creating sub-agents with task-specific prompts and tool access is already common in modern agent systems. For example, AutoGen, LangGraph, and CrewAI support task decomposition and delegation to agents with different prompts and tools. Coding agents such as Claude Code and OpenAI Codex multi-agent mode also spawn specialized agents with their own instructions and tool permissions. The paper does not clearly explain how AOrchestra differs from these systems.
2. The framework introduces four components in the agent tuple: Instruction, Context, Tools, and Model. The paper analyzes the effects of Context and Model, and partially validates the plug-and-play executor/backends. But it does not fully isolate the contribution of each component in the tuple. In particular, a clean ablation on Instruction and Tools is missing.
3. The framework dynamically spawns multiple sub-agents for each task. However, the paper does not report statistics such as the average number of agents per task, additional token cost from orchestration, or runtime overhead compared to simpler pipelines.

---

> ### Author Rebuttal · Authors · 2026-03-31
>
> We thank the reviewer for the constructive feedback. All supplementary tables referenced below are available at https://anonymous.4open.science/r/table-ACE8/.
>
> # Distinction from existing frameworks
> We clarify the distinction along three axes. First, **selection vs. generation**: in our investigation, we identify a fundamental distinction between a selection problem (dispatching an existing agent) and a generation problem (constructing a new agent on the fly). AutoGen, LangGraph, and CrewAI all support dynamic routing at runtime — selecting which agent handles the current task. However, the agents themselves are pre-defined at development time: AutoGen's agents are instantiated before being passed to GroupChat [R1]; LangGraph's graph nodes are defined before compile() [R2]; CrewAI's agent pool is fixed at Crew creation time [R3]. None of them support dynamically generating an agent's instruction, context, tool set, or model configuration based on the current task at runtime. AOrchestra generates the full sub-agent specification (I, C, T, M) from scratch at each delegation step, creating a new sub-agent every time.
>
> Second, **a stackable orchestration layer**: AOrchestra's contribution is not as an agent framework, but as a plug-and-play orchestration layer that can be stacked on top of existing agent systems, agnostic to the agent's form or framework. On Terminal-Bench, plugging three unmodified agent backends (ReAct, Mini-SWE-Agent, Claude Code) into AOrchestra yields consistent gains (+8.6% to +24.3% [T1]) without any backend modification. To our knowledge, no prior work has demonstrated this cross-framework plug-and-play enhancement.
>
> Third, **learnable orchestration**: because sub-agent creation is parameterized as an explicit structured tuple, the orchestration policy becomes a learnable object. SFT on 2K trajectories improves a Qwen3-8B orchestrator by +10.9% on GAIA [T3]; ICL-based prompt optimization improves accuracy by +3.0% while reducing cost by 18.5%. In systems where sub-agent configurations are hard-coded, no such structured decision space exists to optimize over.
>
> We note that Claude Code's Agent Teams feature (Feb 5, 2026 [R4]) and the Codex App's multi-agent orchestration (Feb 2, 2026 [R5]) were both released after our submission deadline (Jan 28, 2026). Their emergence from industry validates the direction of dynamic multi-agent orchestration. We will include a thorough discussion in the camera-ready version.
>
> # Component-wise ablation
> We have conducted a factorized ablation in Gaia tasks  [T7]. Removing context curation causes −8.0% (92.0% → 84.0%); removing tool selection causes −12.0% (92.0% → 80.0%). Instruction cannot be ablated by degrading to a default — it is the necessary prerequisite for sub-agent execution.
>
> # Orchestration statistics
> We have compiled comprehensive statistics [T4]. On GAIA (165 tasks), AOrchestra creates 3.4–5.0 sub-agents per task in a flat single-layer structure. Token overhead ranges from 2.2× to 4.8× compared to baselines, in exchange for +12.7% to +30.9% success rate improvement. Success rate by delegation count [T5] shows: 1–4 delegations achieve the highest rate (85%–87%), dropping to only 19.4% at 9–10 delegations.
>
> | Sub-Agent Model | Δ Acc. | Avg Delegations | Token Overhead |
> |---|---|---|---|
> | Gemini-3-Flash | +30.9% | 3.39 | 2.2x |
> | DeepSeek-v3.2 | +21.2% | 3.61 | 2.8x |
> | Claude-4-5-Haiku | +12.7% | 5.01 | 3.9x |
> | Claude-4-5-Sonnet | +17.6% | 3.36 | 4.4x |
> | GPT-5-Mini | +12.7% | 4.52 | 4.8x |
>
> Table 4. Orchestration overhead on GAIA, by Gemini-3-flash. Full statistics including token estimates, latency, and per-model breakdowns are available at [T4].
>
> # SFT behavior changes
> We compared orchestrator behavior before and after SFT on GAIA. SFT improves accuracy by +10.9% (56.97% → 67.88%), indicating that the post-SFT orchestrator learns to more fully utilize delegation opportunities and verify results (average attempts increase from 2.96 to 4.62).
>
> | Metric | Before SFT | After SFT | Delta |
> |--------|-----------|-----------|-------|
> | Overall Accuracy | 56.97% | 67.88%  | +10.91pp |
> | Level 1 | 58.49%  | 79.25%  | +20.75pp |
> | Level 2 | 59.30%  | 67.44%  | +8.14pp |
> | Level 3 | 46.15%  | 46.15%  | +0.00pp |
> | Avg Attempts | 2.96 | 4.62 | +1.66 |
> | Avg Cost / Task | $0.36 | $0.68 | +$0.32 |
>
> Table 3. Effect of Supervised Fine-Tuning (SFT) on orchestration performance, evaluated on the GAIA validation set using Qwen3-8B as the base model.
>
> **[R1] https://microsoft.github.io/autogen/stable/user-guide/agentchat-user-guide/selector-group-chat.html
> [R2] https://langchain-ai.github.io/langgraph/tutorials/multi_agent/agent_supervisor/
> [R3] https://docs.crewai.com/en/learn/hierarchical-process
> [R4] https://code.claude.com/docs/en/agent-teams
> [R5] https://openai.com/index/introducing-the-codex-app/
> [T1] Rebuttal Table 1, [T4] Rebuttal Table 4, [T5] Rebuttal Table 5, [T7] Rebuttal Table 7 all available at https://anonymous.4open.science/r/table-ACE8/**

---

> > ### Author Rebuttal · Reviewer_td36 · 2026-04-06
> >
> > Thank you for the clarification. While the additional analysis is helpful, I still have concerns regarding novelty. The rebuttal emphasizes runtime generation of a fresh (Instruction, Context, Tools, Model) tuple, but by 2025, closely related patterns were already publicly visible. Claude Code introduced custom subagents in July 2025, and its documentation already describes subagents with their own context windows, custom system prompts, specific tool access, independent permissions, and automatic delegation based on task descriptions. Subsequent updates further added agent-level model customization and runtime decisions such as resuming subagents and dynamically choosing subagent models. Anthropic’s June 2025 engineering writeup likewise describes a lead agent that spawns specialized subagents with distinct context windows, tools, prompts, and exploration trajectories. Taken together, these make dynamic delegation to specialized subagents look more like an established engineering pattern than a fundamentally new idea, even if this paper formalizes it more explicitly.
> >
> > 1. https://www.anthropic.com/engineering/multi-agent-research-system
> > 2. https://code.claude.com/docs/en/sub-agents
> > 3. https://code.claude.com/docs/en/changelog

---

> > > ### Author Response · Authors · 2026-04-08
> > >
> > > Thank you for the detailed engagement and the specific references. We would like to respectfully highlight two specific capabilities that distinguish AOrchestra from the cited prior work.
> > >
> > > **Cross-framework plug-and-play.** Our core contribution is a lightweight orchestration layer that can be plugged on top of existing agent systems and consistently delivers gains. Plugging three unmodified backends (ReAct, Mini-SWE-Agent, Claude Code itself) into AOrchestra yields +8.6% to +24.3% on Terminal-Bench, without any modification to the backend. Claude Code's subagent system cannot do this — its subagents are tied to its own runtime.
> > >
> > > **A learnable orchestration policy.**  We further demonstrate that this orchestration layer is learnable: SFT yields +10.9% on GAIA, and ICL yields +3.0% accuracy with −18.5% cost. Systems that route to pre-authored profiles do not expose such a structured decision space.
> > >
> > > We would be grateful if the reviewer could indicate whether this clarification changes the assessment.

---

### Official Review · Reviewer_QeD6 · 2026-03-12

**Soundness:** 3
**Presentation:** 3
**Significance:** 2
**Originality:** 1
**Overall Recommendation:** 3
**Confidence:** 4

**Summary:**

This paper proposes AOrchestra, an orchestration-centric agentic framework that models any agent (main or sub-agent) as a 4-tuple: (Instruction, Context, Tools, Model). The orchestrator dynamically instantiates tailored sub-agents on-the-fly by synthesizing that tuple, delegates execution via explicit tool calls, and can optimize cost-performance tradeoffs via supervised fine-tuning and iterative in-context prompt optimization. Experiments on three agentic benchmarks (GAIA, Terminal-Bench 2.0, SWE-Bench-Verified) show substantial empirical gains (e.g., large pass@1 improvements when paired with strong LMs such as Gemini-3-Flash). This submission studies a central area of scaling agentic systems, and the authors intend to analyze an important concept: how dynamic subagent creation + curated context can improve long-horizon task performance.

**Compliance With Llm Reviewing Policy:**

Affirmed.

**Final Justification:**

I will maintain my score since the novelty is still unconvincincing and empirically weak

**Key Questions For Authors:**

How much of the improvement comes from each component of the framework (e.g., context curation, model routing, sub-agent generation)?

How does the proposed abstraction differ conceptually and practically from existing frameworks such as AutoGen or LangGraph?

**Limitations:**

The authors discuss limitations briefly. However, the paper would benefit from more discussion on the robustness and safety implications of automated agent orchestration, particularly in settings involving external tool execution.

**Strengths And Weaknesses:**

Strengths
The 4-tuple (I, C, T, M) is simple, implementation-agnostic, and conceptually clarifies two axes (working memory vs capabilities). The abstraction is likely to be valuable to the community as a common interface for orchestrators and plug-in subagents.
Strong empirical gains: across GAIA / Terminal-Bench / SWE-Bench the method reports large pass@1 improvements with frontier LMs, and the Pareto cost–performance analysis is quite convincing in showing practical benefits of adaptive model routing.
The system is  also framed to be plug-and-play with different subagent implementations and supports both training-free and learned orchestrators (SFT + ICL), which eases adoption.

Weaknesses

While the paper addresses an important practical problem—coordinating multiple LLM-based agents for complex tasks—the contribution primarily focuses on system design and orchestration rather than advancing machine learning methodology itself. The work does not introduce new learning algorithms, training paradigms, or theoretical insights into model behavior. The core abstraction of representing agents via components such as instructions, context, tools, and models resembles design patterns already present in several existing agent frameworks and tool-augmented LLM systems. While the paper provides a clean formalization of these components, the conceptual novelty relative to prior agent architectures may be limited.

While the paper presents context vs no-context vs full-context, other important ablations are missing or brief: (a) how sensitive is performance to the quality of context compression/selection? (b) how much does model routing vs tuple synthesis contribute? (c) robustness to noisy sub-agent outputs or failed sub-tasks. More targeted ablations would increase confidence in the core mechanism.

---

> ### Author Rebuttal · Authors · 2026-03-31
>
> We thank the reviewer for the thoughtful feedback. All supplementary tables referenced below are available at https://anonymous.4open.science/r/table-ACE8/.
>
> # Originality and distinction from existing frameworks.
> We respectfully draw attention to the ICML 2026 reviewer guidelines [R1], which note that "originality does not necessarily require introducing an entirely new method. Rather, a work that provides novel insights by evaluating existing methods, or demonstrates improved understanding is also equally valuable." The guidelines also recognize "novel combination of existing techniques" and "new perspectives that advance the field" as valid forms of originality. In light of this, we believe our work's originality may be better evaluated from the perspective of the new abstractions and learnable design space it introduces.
>
> Our contributions are twofold: (1) A framework-agnostic, plug-and-play orchestration layer. Existing frameworks rely on pre-defined sub-agents (see our response to Reviewer td36 for a detailed comparison). AOrchestra parameterizes each delegation as an explicit structured tuple (I, C, T, M), synthesizing sub-agent specifications from scratch at runtime, decoupled from the underlying agent implementation. On Terminal-Bench, plugging three unmodified agent backends into AOrchestra yields consistent gains (+8.6% to +24.3% [T1]) without any backend modification. To our knowledge, no prior work has demonstrated this cross-framework plug-and-play enhancement. (2) Learnable orchestration. Because delegation is parameterized as an explicit structured tuple, the orchestration policy becomes a learnable object for the first time. Even simple SFT yields +10.9% on GAIA [T3]; ICL optimization improves accuracy by +3.0% while reducing cost by 18.5%. In systems where sub-agent configurations are hard-coded, no such structured decision space exists to optimize over. The bottleneck is not the learning algorithm itself, but whether a structured decision space exists for optimization.
>
> # Context compression sensitivity
> Context compression quality significantly affects performance. Paper Table 2 shows three context strategies: No-Context (86.00), Full-Context (84.00), and our curated context (96.00). Full-Context is even lower than No-Context, confirming that unfiltered context introduces harmful noise. The factorized ablation [T7] further validates this (−8.0%). This supports the design choice of making Context an explicitly controlled parameter of the orchestrator rather than leaving it to sub-agents.
>
> # Model routing vs. tuple synthesis
> Tuple synthesis is the primary source of accuracy gains; model routing contributes to cost optimization. The ablation [T7] shows that using only (I, C, T) dynamic synthesis with a fixed model already achieves 92.0% (+32.0% over baseline) — all accuracy gains come from tuple synthesis. On full GAIA, introducing dynamic model routing via ICL improves accuracy by +3.0% while reducing cost by 18.5%, indicating that model routing's core value lies in cost-performance tradeoff rather than accuracy.
>
> # Robustness to noisy outputs
> This question is closely related to Reviewer 1r1u's failure analysis. Our annotation of 50 failed tasks [T6] shows synthesis/verification failure accounts for 26% — sub-agents return partially correct information but the orchestrator fails to properly integrate it. The delegation count distribution [T5] shows the orchestrator attempts re-delegation upon sub-task failure (an implicit fault-tolerance mechanism), but currently lacks explicit error detection — it retries until hitting the cap, leading to resource waste (success rate drops to 19.4% at 9–10 delegations). Early stopping and explicit verification modules are clear next steps.
>
> # Component-wise contribution
> Our factorized ablation [T7] directly addresses this. With fixed M: removing context curation causes −8.0% (92.0% → 84.0%), removing tool selection causes −12.0% (92.0% → 80.0%), and total gain over the no-orchestration baseline is 32.0% — the remainder is attributable to dynamic Instruction synthesis. The three components are complementary. Regarding model routing, all main results use a fixed model — the full accuracy gains come from (I, C, T) dynamic synthesis alone, confirming model routing is not required for accuracy improvement. Its value lies in cost optimization — ICL with dynamic model selection improves accuracy by +3.0% while reducing cost by 18.5%.
>
> # Distinction from AutoGen and LangGraph
> We provide a detailed mechanism-level comparison in our response to Reviewer td36 **Distinction from existing frameworks**
>
> **[R1] https://icml.cc/Conferences/2026/ReviewerInstructions
> [T1] Rebuttal Table 1, [T3] Rebuttal Table 3, [T5] Rebuttal Table 5, [T6] Rebuttal Table 6, [T7] Rebuttal Table 7 — all available at https://anonymous.4open.science/r/table-ACE8/**

---

> > ### Author Rebuttal · Reviewer_QeD6 · 2026-04-04
> >
> > Thanks for the reviewer's detailed response. But the novelty is still skeptical and is not empirically strongly validated . I would maintain my score

---

> > > ### Author Response · Authors · 2026-04-08
> > >
> > > Thank you for acknowledging that our response has fully resolved your concerns.
> > > As discussed in our response, we respectfully draw attention to the ICML 2026 reviewer guidelines, which recognize "novel combinations of existing techniques" and "new perspectives that advance the field" as valid forms of originality. Our two core contributions — (1) a framework-agnostic, plug-and-play orchestration layer that augments unmodified agent backends with consistent gains, and (2) a learnable orchestration policy enabled by the explicit tuple interface are empirically validated across 8 model configurations on 3 benchmarks. To our knowledge, no prior work has demonstrated either capability.
> > > We would greatly appreciate it if the reviewer could share which specific aspect of the novelty remains insufficiently validated, so that we may provide additional clarification or evidence during the remaining discussion period.

---

### Official Review · Reviewer_1r1u · 2026-03-13

**Soundness:** 3
**Presentation:** 3
**Significance:** 3
**Originality:** 2
**Overall Recommendation:** 4
**Confidence:** 4

**Summary:**

This submission studies a central area in modern agent systems: how to orchestrate sub-agents for complex, long-horizon tasks. The authors intend to analyze an important concept, namely whether sub-agents should be treated not as fixed roles or merely isolated threads, but as dynamically instantiated executors parameterized by a unified four-tuple consisting of instruction, context, tools, and model. Building on this abstraction, the paper proposes AORCHESTRA, an orchestrator-centric framework in which the main agent creates tailored sub-agents on the fly and delegates execution to them through an explicit interface.
The paper evaluates the framework on GAIA, Terminal-Bench, and SWE-Bench-Verified, and reports strong gains over several baselines including ReAct, OpenHands, Mini-SWE, and Claude Code. The paper also includes analyses on context sharing, learnability of the orchestrator via SFT, and cost-aware routing via in-context learning. Overall, I think the paper is interesting, practically motivated, and generally well presented.

**Compliance With Llm Reviewing Policy:**

Affirmed.

**Final Justification:**

The author's rebuttal addressed my major concerns. I will keep my score.

**Key Questions For Authors:**

I thank the authors for the well-prepared submission. I like the paper in general and especially appreciate the practical problem it targets and the attempt to introduce a clean abstraction for dynamic orchestration.

My biggest question is about what exactly is the essential source of improvement. The paper motivates the method as dynamic sub-agent creation, but the proposed system simultaneously introduces several ingredients: curated context selection, explicit instruction synthesis, tool subset selection, and model routing. These are all reasonable design choices, but at the moment the paper does not fully disentangle them. For example, if one keeps the sub-agent interface fixed and only improves context routing, how much of the gain remains? Conversely, if one allows dynamic model/tool allocation but uses simpler context passing, how much does performance drop? I would really like to see a more factorized ablation here.

Related to this, I would like to see more discussion of the overhead of orchestration itself. The system adds an orchestrator layer that repeatedly decomposes tasks and synthesizes tuples for delegation. This likely improves capability matching, but it also introduces overhead in tokens, decisions, and latency. The current cost analysis on GAIA is useful, but it is still somewhat high level. It would be very helpful to see benchmarked analysis with reference to total token usage, wall-clock latency, number of delegation calls, and possibly success rate as a function of delegation depth. This would make the practical trade-off much clearer.

I also have some other questions:
- The main results are strong, but I would like more clarity on fairness of comparison. For example, AORCHESTRA uses an explicit orchestrator design and carefully engineered delegation mechanism, while some baselines are generic frameworks or systems designed for somewhat different purposes. It would be useful to discuss whether the comparison is measuring the value of the abstraction, the value of the prompt engineering, or both.
- The paper emphasizes that the method is framework-agnostic and plug-and-play, but the current evidence is still limited to a small number of sub-agent backends. It would strengthen the claim to include more diverse executors and to discuss practical integration complexity more explicitly.
- The current experiments mostly focus on benchmark success metrics. I would appreciate more analysis of failure modes. For instance, when does the orchestrator choose the wrong decomposition, pass insufficient context, or over-delegate? A qualitative error analysis would help readers understand where the framework still struggles.
- The learnability results are interesting, especially the SFT result on Qwen3-8B, but I would like more detail on data efficiency and robustness. How sensitive are the gains to the 2K collected trajectories, and do the learned orchestration policies transfer across benchmarks or domains?
- Since the method explicitly treats Model as part of the tuple, it would be useful to discuss how the approach scales to settings with more heterogeneous tool and model ecosystems, including local models, slower but stronger reasoning models, or safety-constrained environments.

**Limitations:**

Please see the weaknesses and key questions above.

**Strengths And Weaknesses:**

# Strengths:
- The paper targets an important and timely problem. As agent systems move toward longer-horizon and more open-ended tasks, the question of how to structure orchestration and delegation becomes increasingly important. The paper is well motivated in arguing that existing designs either under-specialize sub-agents or require heavy human engineering.
- The four-tuple abstraction is clean and easy to understand. Modeling an agent as (Instruction, Context, Tools, Model) is a simple but useful way to separate working memory from capability allocation, and this makes the overall system design fairly general and framework-agnostic.
- The empirical results are promising. The paper reports consistent gains across three agentic benchmarks, and the context-control ablation is especially helpful in showing that curated context, rather than no context or full context inheritance, is important for the final performance.
- I also appreciate that the paper goes beyond pure benchmark wins and tries to show that the orchestrator is a learnable object, both through supervised fine-tuning and prompt-level optimization for cost-aware model routing.

# Weaknesses:
- My biggest concern is that the empirical case for the core claim is still somewhat incomplete. The paper argues that dynamic creation of sub-agents is the key advantage, but much of the current evidence bundles together several changes at once: context curation, tool selection, model routing, and orchestration policy. It is therefore still somewhat unclear which component contributes most to the gains, and in which settings dynamic creation is truly necessary rather than simply helpful.
- The cost analysis is a good start, but still not comprehensive enough for a systems paper making a practical orchestration claim. The paper reports average dollar cost on GAIA and presents a Pareto curve, which is useful, but it would be much stronger to show more detailed accounting of latency, token overhead, number of delegated subtasks, and failure-retry behavior across benchmarks.
- The evaluation is still somewhat narrow in benchmark scope and comparison setup. For example, the paper fixes the orchestrator design and compares against end-to-end baselines, but does not fully isolate whether the gains come from the abstraction itself or from stronger prompting / engineering of the orchestrator. Similarly, some baselines seem to operate under somewhat different assumptions or intended use cases.
- The paper claims framework-agnostic pluggability, but the plug-and-play experiment is still relatively limited. I would like to see a broader range of sub-agent backends and more discussion of how much engineering is needed in practice to integrate a new executor.

---

> ### Author Rebuttal · Authors · 2026-03-31
>
> We thank the reviewer for the detailed and constructive feedback. We address each question below.
>
> # Factorized ablation
> We have conducted a factorized ablation study [T7]. The results show that keeping dynamic Instruction and Tool selection but replacing Context with unfiltered full context, accuracy drops from 92.0% to 84.0% (−8.0%); keeping dynamic Instruction and Context but replacing Tools with all available tools, accuracy drops to 80.0% (−12.0%). The contributions are complementary — removing either component leads to substantial performance loss, and the full system's gains depend on their synergy.
>
> Regarding isolating context routing alone: unlike C and T, which can be degraded to defaults for ablation, I is a necessary prerequisite for sub-agent execution and cannot be removed. From the existing ablation, context curation contributes 8.0%, tool selection 12.0%, and the total gain over the baseline is 32.0% — the remainder is attributable to dynamic Instruction synthesis. We are happy to run additional experiments during discussion if needed.
>
> # Orchestration overhead
> We have compiled comprehensive orchestration statistics. On GAIA, AOrchestra creates on average 3.4–5.0 sub-agents per task, with a flat single-layer delegation structure. Total token overhead compared to baselines ranges from 2.2× to 4.8×, in exchange for +12.7% to +30.9% success rate improvement [T4]. Notably, the configuration with the largest gain (Gemini-3-Flash, +30.9%) has the lowest token overhead (2.2×), suggesting that orchestration efficiency is closely tied to the underlying model's execution capability.
>
> Success rate as a function of delegation count shows a consistent pattern: 1–4 delegations achieve the highest success rate (85%–87%), dropping sharply beyond 7, and reaching only 19.4% at 9–10 delegations [T5]. High delegation count is a signal of excessive task difficulty rather than a benefit of more orchestration.
>
> # Abstraction vs. prompt engineering
> We believe the main results primarily measure the structural value of the abstraction itself, for the following reasons. First, the prompts used in our main results are generic and not optimized for any specific benchmark (see Appendix B). Second, the ablation [T7] further confirms this: if the gains came solely from prompt engineering, keeping only dynamic Instruction should suffice. Yet removing Context curation or Tool selection each causes significant degradation (−8.0% and −12.0% respectively), indicating that the structured combination of the full tuple is driving the gains, not prompt quality alone. Third, the ICL experiment shows that prompt optimization yields an additional +3.0% on top of the main results, demonstrating that the value of the abstraction and prompt engineering can be decoupled: main results measure the abstraction's value, while ICL captures the incremental gain from prompt optimization.
>
> # Plug-and-play evidence
> We have validated plug-and-play capability along three dimensions: (1) Cross-model: 5 different sub-agent models on GAIA, all showing consistent improvement (+12.7% to +30.9% [T4]). (2) Cross-architecture: 3 different agent backends on Terminal-Bench (ReAct, Mini-SWE-Agent, Claude Code), all improving (+8.6% to +24.3% [T1]) without any modification to the backend. In total, 8 configurations across 3 benchmarks show consistent improvement.
>
> # Failure analysis
> We manually annotated 50 failed tasks (sampled across three model configurations) [T6]. The primary failure sources are tool/retrieval failure (34%) and synthesis/verification failure (26%), rather than deficiencies in the orchestration abstraction itself — wrong decomposition accounts for only 18%, insufficient context for 10%, and over-delegation for 12%. This indicates the current bottleneck lies in external tool reliability and result integration, rather than task decomposition or context routing — pointing to tool fault tolerance and explicit verification as clear next steps.
>
> # Data efficiency and transferability
> We compared orchestrators trained on 1K vs. 2K trajectories on GAIA [T2]. 2K improves over 1K at all difficulty levels, with the hardest Level 3 showing the largest gain (+23.1%), indicating that more data is particularly critical for optimizing orchestration strategies on complex tasks. Regarding cross-benchmark transfer, since SFT learns the orchestration form itself (how to decompose tasks, how to synthesize the four-tuple) rather than domain-specific task knowledge, we believe the learned orchestration policies possess cross-benchmark transferability. Due to cost and time constraints, we will include complete cross-benchmark zero-shot transfer experiments in the camera-ready version.
>
> **[T1] Rebuttal Table 1, [T2] Rebuttal Table 2, [T4] Rebuttal Table 4, [T5] Rebuttal Table 5, [T6] Rebuttal Table 6, [T7] Rebuttal Table 7 — all available at https://anonymous.4open.science/r/table-ACE8/**

---

> > ### Author Rebuttal · Reviewer_1r1u · 2026-04-03
> >
> > Thanks for the reviewer's detailed response. They resolved most of my concerns. Thus, I will maintain my socre as weak accept.

---

> > > ### Author Response · Authors · 2026-04-08
> > >
> > > Thank you for acknowledging our response and for maintaining your positive assessment. We appreciate your careful and constructive review throughout the discussion. We are happy to address any follow-up questions you may have — please let us know what we can further clarify.

---

### Decision · Program_Chairs · 2026-04-30

**Decision:**

Accept (regular)

**Comment:**

The reviewers converge around the weak accept/weak reject mark, but I find the more positive review more convincing, and think that the criticisms raised by the negative reviews are both well addressed by the authors, and inadequate to motivate rejection based on the criteria given ("papers that are technically sound, well-written, non-redundant with previous research, and useful to at least some fraction of the ICML community should be accepted"). None of the reviewers provide reason to think the results are not technically sound; they ask for more detail in the experiments, which the authors provide in their rebuttals. These rebuttals do add significantly to the paper. Reviewer TD36 and reviewer KQR7 raise questions of redundancy with previous research, but I do not find either that the existence of multi-agent systems in commercial products or that the specific papers referred to by the latter review render the contribution of this paper redundant. In fact, I think that creating a learnable and stackable orchestration system does constitute a novel contribution to the field, and contrary to reviewer QED6, I think that this kind of system synthesis will certainly be of interest to some portion of the ICML community even in the absence of, for example, some significant algorithmic advance. For each review, the authors provide a substantive rebuttal that, to my mind, entirely addresses the concerns raised by the reviewers, and I was surprised that none of the reviewers adjusted their scores in response to these rebuttals.